# Is Data Shapley Not Better than Random in Data Selection? Ask NASH

**Xiao Tian** [* 1 2] **Jue Fan** [* 1 2] **Rachael Hwee Ling Sim** [1] **Zixuan Wang** [1] **Nancy F. Chen** [2] **Bryan Kian Hsiang Low** [1]

## Abstract

Data selection studies the problem of identifying high-quality subsets of training data. While some existing works have considered selecting the subset of data with top-$m$ Data Shapley or other semivalues as they account for the interaction among every subset of data, other works argue that Data Shapley can sometimes perform ineffectively in practice and select subsets that are *no better than random*. This raises the questions: **(I)** *Are there certain "Shapley-informative" settings where Data Shapley consistently works well?* **(II)** *Can we strategically utilize these settings to select high-quality subsets consistently and efficiently?* In this paper, we propose a novel data selection framework, **NASH** (**N**on-linear **A**ggregation of **SH**apley-informative components), which **(I)** decomposes the target utility function (e.g., validation accuracy) into simpler, Shapley-informative component functions, and selects data by optimizing an objective that **(II)** aggregates these components non-linearly. We demonstrate that NASH substantially boosts the effectiveness of Shapley/semivalue-based data selection with minimal additional runtime cost. Our implementation is provided in https://github.com/snoidetx/nash.

## 1 Introduction

As the scale of machine learning (ML) grows larger, *data selection* (Iyer et al., 2021; Albalak et al., 2024), which aims **to select the best subset of training data of a limited size** $m$, has become increasingly important. For example, the ML model owner may have a limited budget to purchase data or a limited storage to store data for training; the raw feasible set may contain harmful data that degrade model utility.

---
[*]Equal contribution [1]Department of Computer Science, National University of Singapore, Singapore [2]Agency for Science, Technology and Research (A*STAR), Singapore. Correspondence to: Bryan Kian Hsiang Low <lowkh@comp.nus.edu.sg>.

*Proceedings of the 43rd International Conference on Machine Learning*, Seoul, South Korea. PMLR 306, 2026. Copyright 2026 by the author(s).

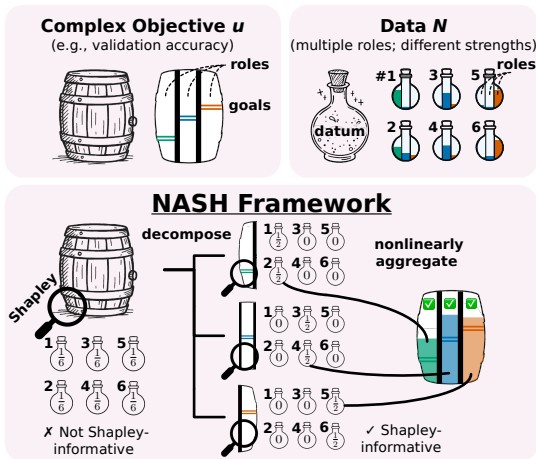

*Figure 1.* **Overview of NASH data selection framework.** When the utility function (e.g., validation accuracy $u_V$) is complex and involves different *roles* of data (e.g., 3 roles in the figure), Shapley values fail to inform on data with different strengths (e.g., different columns of data in the figure), and thus may perform badly. To address this, our NASH first decomposes $u_V$ to simpler, fewer-role components where Shapley values are informative (i.e., each compartment in the figure). It then aggregates these *Shapley-informative components* non-linearly to account for the interaction among selected data.

Specifically, the utility of a selected subset can be quantified through some utility function $u$, such as the commonly used *validation accuracy* $u_V$ on a held-out validation set $V$.

However, **optimizing $u_V$ is computationally challenging**. Bruteforce search over all candidate subsets requires exponential number of *evaluations* of $u_V$ (note that the evaluation of $u_V$ on each candidate subset requires one round of model training and evaluation). Even if $u_V$ is considered (approximately) monotone submodular, the greedy algorithm (i.e., selecting the datum that maximally improves $u_V$ at each round) requires polynomial number of evaluations of $u_V$. To address this challenge, a common data selection heuristic is the ***top-$m$ heuristic***: **to assign each training datum a numerical score computed using *data valuation* methods (Sim et al., 2022), and select the subset of training data with top-$m$ scores**.

*Data Shapley* (Ghorbani & Zou, 2019; Jia et al., 2019a) and its generalization, *semivalues* (Dubey et al., 1981; Kwon & Zou, 2022; Wang & Jia, 2023), are well-known data val-

uation methods and thus also commonly applied to data selection tasks. Sec. 2 gives a detailed review of these methods. An important principle of Data Shapley and semivalues is that (†) **each datum's worth depends on other data present in the training set**. Thus, each datum $i$'s valuation score should incorporate its (marginal) contribution to every subset $S$ of other training data (i.e., $u(S \cup \{i\}) - u(S)$). This principle is particularly relevant to data selection, because it is sensible that one should not *myopically* view each datum as working independently when trying to select a subset that jointly works well. Hence, prior works (Ghorbani & Zou, 2019; Kwon & Zou, 2022; Wang et al., 2024a) have empirically demonstrated the effectiveness of Data Shapley and semivalues (using $u_V$ as utility function) in data selection, compared with other methods that do not consider interactions such as *leave-one-out (LOO) values* and *influence functions* (Koh & Liang, 2017). Nonetheless, recent works (Wang et al., 2024c; Chi et al., 2025) show that **selecting via top-$m$ heuristic with Data Shapley based on $u_V$ may sometimes result in an under-performing subset**, which can be even worse than random selection (Sec. 2).

The above paradox implies that Data Shapley is only effective for *certain* utility functions $u$. **Can we make Data Shapley consistently effective in ML data selection (e.g., using $u_V$ as utility function)?** Intuitively, selecting with Data Shapley works well when each subset's sum of Shapley values is *indicative* of its actual utility. We formalize this *Shapley-informativeness* property and analyze why *arbitrary* utility functions $u$, as well as specific utility functions $u_V$ (induced by ML), are not Shapley-informative. Thus, we ask the question: ***Is there any Shapley-informative ML utility functions** such that each subset's sum of Shapley values is well indicative of its actual utility?* We show that simpler utility functions, such as *prediction correctness* for a single validation datum, are provably Shapley-informative under reasonable assumptions. We name such simpler functions *Shapley-informative components*, which will serve as building blocks for our proposed data selection framework. This leads to the next question: *Given Data Shapley's advantages (†), **can we strategically utilize the Shapley-informative components to consistently achieve effective data selection and optimize** $u_V$?* At first glance, readers may think of maximizing the *average* of these components. However, this *linear* aggregation recovers the flawed vanilla Data Shapley. Therefore, we present a novel data selection framework, **NASH** (**N**on-linear **A**ggregation of **SH**apley-informative components), which uses a sum of concave functions to aggregate Shapley-informative components. Fig. 1 gives an overview of NASH. We also explain how the NASH objective can be optimized effectively and efficiently using the greedy algorithm, thus introducing minimal additional runtime cost to current Shapley/semivalue-based data selection methods. **Notably, existing works on *efficient* approximation of Shapley values/semivalues** (e.g., Ghorbani & Zou (2019); Li & Yu (2024); Wang et al. (2024a)) **can be freely plugged into NASH and become *effective* in data selection**, which further validates the practicality of NASH.

Our contributions are summarized as follows:

- We formalize the *Shapley-informativeness* property that ensures the effectiveness of Data Shapley for data selection, and analyze why commonly used utility functions fail to satisfy it thus making Data Shapley ineffective (Sec. 3.1);
- We show that simpler utility functions, such as *prediction correctness* for a single validation datum, are provably Shapley-informative (Sec. 3.2);
- We propose a novel data selection framework NASH based on the above Shapley-informative components, which consistently selects data effectively and efficiently (Sec. 3.3);
- We empirically demonstrate that NASH substantially boosts the effectiveness of Data Shapley and semivalues using various datasets and models (Sec. 4).

**Significance to the research community.** Due to the inconsistent performance of Data Shapley/semivalues in data selection, existing works have only demonstrated that Data Shapley is beneficial on limited datasets and settings (e.g., injecting noise to the datasets). NASH largely eliminates this restriction and shows that Data Shapley/semivalues can be consistently reliable in data selection and have robust performances across datasets/settings, which would hence benefit future related works (e.g., on improving the approximation efficiency).

## 2  Background

Let $N$ be the *feasible set* of data (to be selected from) with $|N| = n$. Let $u : 2^N \to \mathbb{R}$ be a *utility function* that measures the utility of ML model trained using subset $M \subseteq N$, $u(M)$. A commonly used utility function is $u_V$, the *validation accuracy* on validation set $V$. Let the selection size be $m$. We focus on the following conventional formulation of the data selection problem (Wang et al., 2024b):

$$M^* \coloneqq \arg\max_{M \subseteq N; |M| \le m} u(M) . \tag{1}$$

A natural way to identify high-quality subsets of training data $M \subseteq N$ is to quantify the value of each datum and select the ones with highest values (i.e., top-$m$ heuristic in Sec. 1). To achieve this, *data valuation* works (Koh & Liang, 2017; Sim et al., 2022) assign each datum a numerical score representing its worth. *Data Shapley* (Ghorbani & Zou, 2019; Jia et al., 2019a) is a principled data valuation method which models ML as a *cooperative game* among all data (as players), such that each *coalition* (subset) $S \subseteq N$ of data can achieve the model utility $u(S)$. In particular, Data

Shapley utilizes the *Shapley value* (Shapley, 1953) which computes the valuation score of each datum $i$ by aggregating its *marginal contribution* to every coalition $S$ formed by other data, $\Delta_u(i|S) := u(S \cup \{i\}) - u(S)$. Formally,

**Definition 2.1** (Shapley Value). The *Shapley value* of a datum $i \in N$ with regards to (w.r.t.) utility function $u$ is

$$\phi_i(u) := \sum_{S \subseteq N \setminus \{i\}} \frac{1}{n} \cdot \Delta_u(i|S) / \binom{n-1}{|S|} . \tag{2}$$

Data Shapley has two major advantages in the context of data selection. Firstly, it is **non-myopic** as Eq. (2) accounts for the interaction/synergy within every subset of data, which is important to selecting a subset $M$ that *jointly* gives a high utility. Secondly, it is **equitable** such that each datum is valued proportionally to its actual contribution. Equitability is characterized by a list of axioms which ensures data are not overvalued or undervalued, such as

**[INT]** *Interchangeability*: If data $i$ and $j$ are interchangeable w.r.t. utility function $u$ (i.e., $u(S \cup \{i\}) = u(S \cup \{j\})$ for all $S \subseteq N \setminus \{i\}$), then $\phi_i(u) = \phi_j(u)$.

App. B.1 gives the full list of equitability axioms. Notably, Data Shapley is the only data valuation method that satisfies all these axioms while enforcing the sum of all data's valuation scores to be $u(N)$.[1] This additional constraint can sometimes be less important and removed, which gives the relaxation of Shapley value, *semivalues* (Dubey et al., 1981): $\varphi_i(u) := \sum_{S \subseteq N \setminus \{i\}} w_{|S|} \cdot \Delta_u(i|S) / \binom{n-1}{|S|}$, where $w_{|S|}$'s are non-negative *weighting coefficients* such that $\sum_{s=0}^{n-1} w_s = 1$. Common semivalues include *Beta Shapley* (Kwon & Zou, 2022) and *Data Banzhaf* (Wang & Jia, 2023) (see App. B.2 for more details).

Exact computation of Shapley values and semivalues requires evaluation of $u(S)$ for every $S \subseteq N$, which is challenging. Fortunately, existing works on efficient approximation of Shapley values and semivalues (Sec. 5) greatly improve their practicality and complement this work.

**Data Shapley for data selection.** The aforementioned advantages make Data Shapley and semivalues promising approaches for data selection. Prior works (Ghorbani & Zou, 2019; Tang et al., 2021; Wang et al., 2024a) empirically demonstrate that Data Shapley w.r.t. validation accuracy $u_V$ gives stronger performance than other methods that do not possess these advantages such as *leave-one-out (LOO) values* and its approximation, the *influence function* (Koh & Liang, 2017), which value each datum based on the effect of removing it from the feasible set and hence do not consider the interaction within each subset of data. App. B.5 gives a review of these methods. However, recent works (Kwon & Zou, 2022; Wang et al., 2024c) identify

that Data Shapley w.r.t. validation accuracy $u_V$ sometimes performs less competitively and can even be *no better than random*. This leads to the question: *When does Data Shapley perform well for data selection?* Under top-$m$ heuristic, Wang et al. (2024c) discovers that Data Shapley works well on heterogeneous datasets (i.e., contains low-quality/noisy data) or w.r.t. *monotonically transformed modular (MTM) functions*; Chi et al. (2025) shows that Data Shapley works well on utility functions with small curvature (i.e., close to modular functions) (see App. B.3 for a summary of the reasoning). Unfortunately, the actual utility function (e.g., $u_V$) may not always coincide with these settings and thus Data Shapley may give inconsistent performances. To address this challenge and make Data Shapley consistently effective, in this work, we do not adopt top-$m$ heuristic. Instead, we decompose the complex objective $u_V$ into simpler utility functions where Data Shapley gives desirable behaviors (i.e., *Shapley-informativeness* in Sec. 3.1). We then aggregate these Shapley-informative components non-linearly to select a high-quality subset $M$.

## 3 Methodology

In Sec. 3.1, we formulate the *Shapley-informativeness* property for utility functions, a desirable behavior for Data Shapley to be effective data selection. We then motivate why the commonly used validation accuracy $u_V$ is not Shapley-informative and thus might cause ineffective data selection. Based on these motivations, in Sec. 3.2, we introduce a type of provably Shapley-informative utility functions, which will serve as building blocks (i.e., *Shapley-informative components*) for our proposed method. In Sec. 3.3, we formulate our NASH objective that properly aggregates these Shapley-informative components and propose an effective and efficient algorithm to optimize it.

### 3.1 Shapley-Informativeness

Data valuation methods such as Data Shapley assess the quality of data using valuation scores. As motivated in Sec. 1, for such methods to work in data selection, it is desirable that a set of high-quality data should in general result in high utility, and vice versa. This requires each subset $M \subseteq N$'s sum of Shapley values, $\hat{u}(M)$, to be indicative of its actual utility $u(M)$. To formalize this idea, we define the following *Shapley-informativeness* property probabilistically:

**Definition 3.1** (Shapley-Informativeness). A utility function $u$ is $(\epsilon, \delta)$-*Shapley-informative at size $m$ under mapping* $f : \mathbb{R} \to \mathbb{R}$ if for a size-$m$ subset $\mathcal{M} \subseteq N$ drawn uniformly at random,

$$\Pr[|u(\mathcal{M}) - f(\hat{u}(\mathcal{M}))| \le \epsilon] \ge 1 - \delta, \tag{3}$$

where $\hat{u}(\mathcal{M}) := \sum_{i \in \mathcal{M}} \phi_i(u)$ is the sum of Shapley values

---

[1] This constraint means that the overall utility $u(N)$ is attributed/broken down to every single datum in $N$.

in $\mathcal{M}$.

Intuitively, if $u$ is Shapley-informative, then if one computes the Shapley values and knows a proper mapping $f$, one can recover the actual utility with high probability and thus select desirable subsets. As an extreme example, a *modular* utility function is $(0,0)$-Shapley-informative at any size under the identity mapping $f(x) = x$ (App. C.1 gives more simple examples).

*Remark.* An *arbitrary* utility function $u : 2^N \to \mathbb{R}$ (not necessarily in the context of ML) is in general unlikely to be Shapley-informative under any mapping $f$. This is because each $u$ corresponds to a unique size-$2^n$ vector $\mathcal{G}_u \in \mathbb{R}^{2^n}$ such that each entry of $\mathcal{G}_u$ corresponds to a unique coalition $S \subseteq N$ and equates to $u(S)$. Data Shapley is a linear mapping from $\mathcal{G}_u \in \mathbb{R}^{2^n}$ to $\phi_u \in \mathbb{R}^n$ (i.e., the $n$ Shapley values $\phi_i(u)$) of rank $n$ and hence nullity $2^n - n - 1$.[2] Hence, for any utility function $u$ and corresponding $\mathcal{G}_u$, adding any linear combination of basis vectors from the null space of Data Shapley mapping would result in the same Shapley values $\phi_u$. Thus, there is no mapping $f$ that can recover all original $u$ from the Shapley values $\phi_u$ or its induced $\hat{u}(M)$'s (even when a small probability of violation $\delta$ is allowed). For instance, App. C.2 gives an example where utility functions with different desirable subsets have the same Shapley value vector (and thus one cannot identify desirable subsets through any mapping $f$).

It is our view that **in the context of ML and Data Shapley, the utility function** $u : 2^N \to \mathbb{R}$ **is *not* arbitrary**, but always induced by ML task-specific datasets and models. Hence, the negative conclusion about arbitrary utility functions in the above remark does not necessarily carry over. Indeed, data selection with Data Shapley consistently demonstrates good performance on *certain* ML tasks (Ghorbani & Zou, 2019; Kwon & Zou, 2022; Wang et al., 2024a), which implies that *some* **ML-induced utility functions** $u$ **possess certain structures or properties that make them Shapley-informative**. We can thus identify and exploit such functions to select high-quality subsets.

To envision what ML-induced utility functions are Shapley-informative, we first examine the commonly used utility function, *validation accuracy* $u_V$. Unfortunately, $u_V$ is unlikely to be Shapley-informative: As shown in Fig. 3a, the (linear) sum of Shapley values $\hat{u}_V(M)$ is not indicative of the actual utility $u_V(M)$ (this observation is also supported by prior works (Wang et al., 2024c; Chi et al., 2025)). We identify the key reason as follows: **Complex utility functions such as** $u_V$ **involve multiple distinct *roles***, and **each feasible datum** $i \in N$ **has its unique *strength* in a certain set of roles**. For example, $u_V$ averages across validation data from different parts of the data space (e.g., different classes or subpopulations),

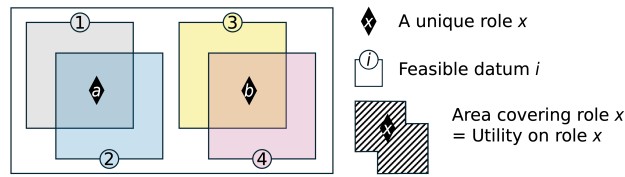

*Figure 2.* **Complex utility functions involving multiple roles (e.g., $u_V$) is not Shapley-informative.** In this toy illustration, all data have the same Shapley value. Yet, if ① has been selected, ③ would contribute more than ② as role $b$ is uncovered.

which correspond to different sets of roles and require training data with different strengths (e.g., from every class or subpopulation). This raises two serious problems that make $u_V$ not Shapley-informative (illustrated in Fig. 2):

**P1** **Each datum** $i$**'s Shapley value** $\phi_i(u_V)$ **does not capture its strength.** For example, consider 2 training data $i, j$ with equal Shapley values $\phi_i(u_V) = \phi_j(u_V)$ but different strengths (e.g., ② and ③ cover different areas in Fig. 2). Although $i$ and $j$ behave differently, any data selection method based solely on the Shapley values $\phi_{u_V}$ would treat $i$ and $j$ interchangeably. However, data with different strengths should not be treated interchangeably. For example, data strong at similar roles may weaken each other's contributions, while data with complementary strengths may amplify them (Hu et al., 2024). Hence, any data selection method based solely on $\phi_{u_V}$ cannot capture these effects (i.e., $\hat{u}_V$ is not indicative of $u_V$) and may select data that are no better than random.

**P2** **The objective** $u_V(M)$ **aggregating different roles cannot be represented by a linear aggregation (e.g., sum) of values in** $M$**.** Linearity implies that selection of some data does not affect selection of other data (since the values do not interact). This is undesirable when maximizing $u_V$ because a datum has different instead of same contribution to different coalitions depending on how it interacts with data within them. For example, if some roles/parts of the data space have already been predicted correctly (e.g., ① and Role 1 in Fig. 2), the new data to be added should focus on the uncovered roles/underrepresented parts of the data space (e.g., Role 2 in Fig. 2) instead of the covered ones. In practice, this issue causes Data Shapley to select duplicated data or data from only a single class as their values can be similarly high and lie inside the top-$m$ data. Thus, a non-linear aggregation should be used.

To address **P1** while retaining the advantages of Data Shapley (Sec. 2), in Sec. 3.2 we break down $u_V$ to simpler (component) utility functions $u_v$'s such that each $u_v$ corresponds to a single role (but each training datum has different contribution to this role) or a set of similar roles. Intuitively, each

---

[2] $-1$ here since $u(\emptyset) = 0$ by convention.

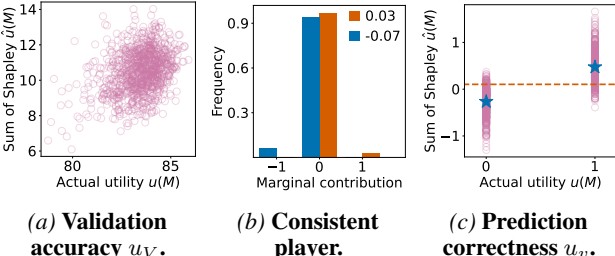

*(a)* **Validation accuracy** $u_V$.

*(b)* **Consistent player.**

*(c)* **Prediction correctness** $u_v$.

*Figure 3.* **Illustration of Shapley-Informativeness.** 3a shows that validation accuracy is not Shapley-informative since the same sum of Shapley values $\hat{u}$ can correspond to a range of actual utility $u$; 3b shows the distribution of marginal contributions when two data with different goodness (can be seen from their Shapley values) join different subsets, and they both demonstrate consistency; 3c shows that prediction correctness on a single validation datum is Shapley-informative with regards to a proper threshold function.

component $u_v$ would be Shapley-informative, which we theoretically prove. To address **P2**, in Sec. 3.3 we propose a non-linear aggregation of Shapley-informative components $u_v$'s as our optimization objective (i.e., proxy to $u_V$) and show that it can be optimized effectively and efficiently.

### 3.2 Shapley-Informative Components

In Sec. 3.1, we motivate that **a utility function $u$ involving a single role is likely to be Shapley-informative**. Yet, for the complex objective $u_V$ that involves model training and evaluation, it is hard to know exactly what the *single* roles are or how to break down $u_V$ to such roles (discussed further in App. C.3). However, a natural way to decompose $u_V$ into *simpler* components (that is likely to involve much fewer roles) is to exploit each individual validation datum $v \in V$ and define the utility function $u_v : 2^N \to \{0, 1\}$ such that $u_v(S) = 1$ when the ML model trained on coalition $S$ predicts validation datum $v$ correctly (i.e., *prediction correctness*). Note that $u_V(S) = (1/|V|) \sum_{v \in V} u_v(S)$.

*How is the new utility function $u_v$ simpler?* Intuitively, each training datum's contribution to a validation datum $v$ is clear-cut: it is either good, harmful, or irrelevant to $v$. The typical empirical behavior is shown in Fig. 3b, where a "good" player indicated by a positive Shapley value either improves $u_v$ from 0 to 1 or makes no change, whereas a "bad" player indicated by a negative Shapley value either decreases $u_v$ from 1 to 0 or makes no change.

Since $u_v$ is simpler and involves much fewer roles, we will now show that $u_v$ **is provably Shapley-informative**. In Sec. 3.1 we have explained why an arbitrary utility function is in general not Shapley-informative, thus suitable assumptions are needed to model $u_v$. Based on our earlier observations (e.g., Fig. 3b), we make the following Consistent Player assumption about $u_v$:

**Assumption 3.2** (Consistent player)**.** Each training datum

contributes *consistently* w.r.t. $u_v$: A good datum is always good (contributes either 0 or 1, but never $-1$), and vice versa. Formally, each datum $i \in N$ is associated with a *goodness indicator* $g_i \in \{+1, -1\}$ such that

$$\forall S \subseteq N \setminus \{i\} \quad [\Delta_{u_v}(i|S) \in \{0, g_i\}]. \tag{4}$$

In App. C.4, we show that empirically Assump. 3.2 has a low violation rate. Thus, the assumption is reasonable and necessary to establish a theoretical framework that designs and analyzes Shapley-informative utility functions (e.g., Prop. 3.3). Note that it is more flexible than existing assumptions such as the MTM assumption (Wang et al., 2024c) (in Sec. 2) which requires the ranking of the contribution of data points to be the same across all coalitions.

In Fig. 3c, we plot the graph of sums of Shapley values $\hat{u}(M)$ against actual utilities $u(M)$ for different coalitions $M$. It is clear that $\hat{u}(M)$ is indicative of $u(M)$: if $\hat{u}(M)$ is larger than a proper threshold (dashed line in Fig. 3c), then $u(M) = 1$ with high probability, vice versa. Theoretically, we show that this is not a coincidence and is a result of the Consistent Player assumption (Assump. 3.2):

**Proposition 3.3** (Informal)**.** *Suppose $u_v$ satisfies Assump. 3.2. Let $f_\tau$ be the threshold function (i.e., $f_\tau(\hat{u}_v(S)) = \mathbb{I}[\hat{u}_v(S) \geq \tau]$). Then at any selection size $m$, there exists $\tau \leq 1$ such that $u_v$ is $(0, \delta)$-Shapley informative under mapping $f_\tau$.*

The formal proposition (including the closed form of $\delta$) and proof are given in App. C.6. To summarize, we first show that Assump. 3.2 leads to Lem. C.1 (in App. C.5), which states that the expected sum of Shapley values for subset $\mathcal{M}_0$ (whose actual utility $u(\mathcal{M}_0)$ is 0) is smaller than the expected sum of Shapley values for subset $\mathcal{M}_1$ (whose actual utility $u(\mathcal{M}_1)$ is 1) (let $\gamma$ denote their gap). This can also be empirically observed in Fig. 3c where the two expectations are shown as ★. Then, by showing how each subset $M$'s sum of Shapley values concentrates around its corresponding expectation, we can bound the probability that the sum of Shapley values falls outside the threshold. The bound given in Prop. 3.3 is realistic. For example, in App. C.6, we give an example with $\delta = 3.6e-3$ at selection size 400 out of 2000 data.

*Remark.* Prior works (Ghorbani & Zou, 2019; Wang et al., 2024c) demonstrate that Data Shapley w.r.t. validation accuracy $u_V$ works well for heterogeneous-quality datasets (e.g., with corrupted data). This aligns with our analysis because Assump. 3.2 is partially satisfied: The corrupted data are consistently bad, whereas the uncorrupted data are generally better. Hence, Data Shapley can distinguish the corrupted data from the uncorrupted ones well.

*Remark.* The time taken for computing the Shapley value vectors $\phi_{u_v}$'s for all validation data $v \in V$ is the same

as the time taken for computing the Shapley value vector $\phi_{u_V}$ (validation accuracy), because validation accuracy is computed by averaging every prediction correctness. Hence, **computing all $\phi_{u_v}$'s does not incur extra time cost**.

Given the Shapley-informative components $u_v$'s, the next question becomes: *How do we aggregate these $u_v$'s to get the highest validation accuracy $u_V$ on the entire validation set?* In the following section, we explain why and how a non-linear aggregation should be made.

### 3.3 NASH: Non-Linear Aggregation of Shapley-Informative Components

Given that simpler utility functions (e.g., prediction correctness $u_v$) are Shapley-informative (i.e., sum of Shapley values $\hat{u}_v$ indicates $u_v$ well), *how should one properly aggregate these Shapley-informative components to optimize the true, complex objective (e.g., validation accuracy $u_V$)?* At first glance, readers may think of maximizing the *average/sum* of all $\hat{u}_v$'s. However, we have argued in Sec. 3.1 **P2** that the true objective $u_V$, which aggregates different roles, cannot be represented as a linear aggregation. Indeed, by Linearity of the Shapley value (App. B.1), $(1/|V|)\sum_{v \in V} \hat{u}_v(M) = (1/|V|)\sum_{v \in V} \sum_{i \in M} \phi_i(u_v) = \sum_{i \in M} \phi_i\left((1/|V|)\sum_{v \in V} u_v\right) = \hat{u}_V(M)$. Maximizing the average of all $\hat{u}_v$'s is equivalently maximizing the sum of Shapley values w.r.t. validation accuracy $u_V$ (i.e., top-$m$ Shapley values), which is ineffective.

Now recall Prop. 3.3. Suppose we know the threshold $\tau_v$ corresponding to each validation datum $v \in V$. Then, if the sum of Shapley values $\hat{u}_v(M) \geq \tau_v$, then $u_v(M) = 1$ (and $v$ is predicted correctly) with high probability. Optimizing validation accuracy $u_V$ is equivalently optimizing the number of validation points $v$ on which the threshold $\tau_v$ is met:

$$\max_{\substack{M \subseteq N \\ |M|=m}} \quad \sum_{v \in V} \mathbb{I}\left[\hat{u}_v(M) \geq \tau_v\right], \tag{5}$$

where $\mathbb{I}[\cdot]$ is the *indicator function* which evaluates to 1 if condition $\cdot$ is met and 0 otherwise. However, estimating each validation datum $v$'s threshold $\tau_v$ is computationally challenging, as one needs to gather a large number of samples $M$ for which $u_v(M) = 0$ and 1 respectively for each validation datum $v \in V$. *Can we relax Obj. (5) to one that does not depend on the exact $\tau_v$'s so that it is computationally feasible?*

Let $\mathcal{T}$ be the random variable denoting the threshold value $\tau_v$ of each validation datum $v$ drawn from $V$. Let $p_{\mathcal{T}}$ denote the distribution of $\mathcal{T}$ and $F_{\mathcal{T}}$ denote the cumulative distribution function (c.d.f.) of $p_{\mathcal{T}}$. Since the exact $\tau_v$'s are unknown, one can instead maximize the expectation of Obj. (5). The expectation of the indicator $\mathbb{I}\left[\hat{u}_v(M) \geq \tau_v\right]$ is $\Pr[\hat{u}_v(M) \geq \mathcal{T}] = F_{\mathcal{T}}(\hat{u}_v(M))$. Thus, the expected

objective is equivalent to the following:

**Definition 3.4** (NASH Objective[3])**.** Given Shapley informative components $u_v$'s and the corresponding Shapley values, the NASH *objective* is defined as

$$\max_{\substack{M \subseteq N \\ |M|=m}} \quad \sum_{v \in V} F_{\mathcal{T}}\left(\hat{u}_v(M)\right). \tag{6}$$

The next step is to set $F_{\mathcal{T}}$ and optimize the NASH objective.

**Choice of $F_{\mathcal{T}}$.** The closed form of $F_{\mathcal{T}}$ can be inspired from the *learning curves* (Viering & Loog, 2022), which describe how model utility changes as one scales the number of training data $m$ in the *general* ML setting (where the training set $\mathcal{M} \subseteq N$ with $|\mathcal{M}| = m$ is selected uniformly at random). It is known that such curves can be well fitted by concave functions such as an exponential law (i.e., $u(\mathcal{M}) \approx u_{\max} - \alpha \exp(-\beta m)$) or a power law (i.e., $u(\mathcal{M}) \approx u_{\max} - \alpha m^{-\beta}$) (Viering & Loog, 2022). In this random selection setting, the expected sum of Shapley values $\mathbb{E}[\hat{u}_v(\mathcal{M})]$ scales *linearly* with $m$: $\mathbb{E}[\hat{u}_v(\mathcal{M})] = (m/n) \cdot \hat{u}_v(N) = m/n$, because each feasible datum is equally likely to be included in $\mathcal{M}$. Thus, since the learning curves inform how model utility increases as $m$ increases, they also inform how likely the threshold $\mathcal{T}$ is met as the (expected) $\hat{u}_v(\mathcal{M})$ increases. For example, given the exponential law and $m$, we can set $F_{\mathcal{T}}$ such that it has the same value as the law when $\hat{u}_v(\mathcal{M})$ takes on the expected value $(m/n) \cdot \hat{u}_v(N)$ under random selection, i.e., $F_{\mathcal{T}}(x) := u_{\max} - \alpha \exp(-\lambda x)$ for $\lambda := \beta n / \hat{u}_v(N)$.[4] In App. D.5.1, we compare across $F_{\mathcal{T}}$'s based on different laws and the above exponential law gives the best performance.

At a high level, the NASH objective (6) addresses **P2** (recall that **P1** is addressed by Sec. 3.2): When $F_{\tau}$ is concave and increasing, it would be increased by a larger extent by a datum that contributes to uncovered roles/validation data that are predicted wrongly (i.e., thresholds not yet met). Thus, a

---

**Algorithm 1 The NASH algorithm.** We consider the exponential law-based $F_{\mathcal{T}}$ with one hyperparameter $\lambda$ (see Footnote 4 for why only the $-\exp$ term remains).

**Input:** Feasible set $N$, validation set $V$, size $m$, lambda $\lambda$.
**Output:** Selected set $M$.
1: Pre-compute the Shapley value vectors $\phi_{u_v}$ for all $v \in V$
2: $M \leftarrow \emptyset$
3: **for** round $t = 1, 2, \cdots, m$ **do**
4: $\quad i^* \leftarrow \arg\max_{i \in N \setminus M} \sum_{v \in V} -\exp(-\lambda \hat{u}_v(M \cup \{i\}))$
5: $\quad M \leftarrow M \cup \{i^*\}$
6: **end for**
7: **return** selected set $M$

---

[3]Fun fact: When $F_{\mathcal{T}}$ is set based on the logarithmic law, NASH optimizes the *Nash welfare* (Nash et al., 1950) across all validation data $v \in V$.

[4]Since Obj. (6) is a summation of $F_{\mathcal{T}}$'s, the values of $u_{\max}$ and $\alpha$ do not matter and one only needs to tune $\lambda$ in practice.

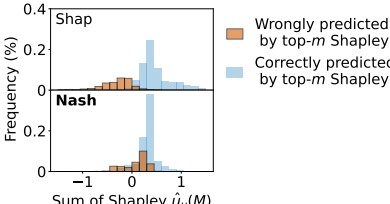

*Figure 4.* **Qualitative analysis of subsets selected by Data Shapley and NASH.** We use a $40\%$ subset of our **PO-LR** setting (see Sec. 4) and show histograms of $\hat{u}_v(M)$ on different validation data $v \in V$ or roles.

datum would have different contributions to different sets depending on the strengths of data in these sets. Fig. 4 provides an empirical example that further illustrates how NASH addresses **P2**: (1) From the histogram of Shapley (top), validation data with higher sum of Shapley values $\hat{u}_v(M)$ (so more likely $\hat{u}_v(M) \geq \tau_v$) are more likely to be correctly predicted, vice versa; (2) Data Shapley accumulates unnecessarily high sum of Shapley values $\hat{u}_v(M)$'s on some validation data (represented by the heavy right tail); (3) the subset selected by NASH (bottom) results in a higher sum of Shapley values $\hat{u}_v(M)$ on the validation data that are wrongly predicted by the subset selected by vanilla Shapley, making them more likely to exceed the thresholds and become correctly predicted. This explains why NASH accounts for more roles through a non-linear aggregation and leads to a much better performance.

**Optimization of NASH objective (6).** Alg. 1 gives the NASH algorithm that maximizes the NASH objective. Specifically, the NASH objective (6) can be maximized effectively and efficiently using the greedy algorithm (i.e., adding the datum that increases Obj. (6) by the most at each round). When all Shapley values are non-negative, Obj. (6) is monotone submodular because it is a sum of concave over modular functions and the greedy algorithm gives a $(1 - 1/e)$-approximation (Nemhauser et al., 1978). For datasets where not all Shapley values are non-negative, we notice that the main source of non-submodularity is the harmful/noisy data (indicated by very low $\phi_i(u_V)$), which are not desirable. For example, in our **WD-LR** experiment, the top $80\%$ feasible data gives a submodularity ratio $\approx 0.85$. Note that the excluded harmful data will not be preferred by the greedy algorithm. In App. C.7, we empirically demonstrate the effectiveness of the greedy algorithm.

The NASH algorithm introduces minimal additional runtime cost to existing methods that compute/approximate the Shapley values/semivalues. Calculation of all $\phi_i(u_v)$'s incurs almost no additional runtime to the traditional $\phi_i(u_V)$ because one already obtains all $u_v(S)$'s to calculate their average $u_V(S)$ for each sampled subset $S$. The greedy algorithm is also efficient in our case since all operations can be vectorized. For example, selecting 10,000 out of

20,000 training data w.r.t. 5,000 validation data takes only $\sim 53.4$ seconds wallclock runtime. We also demonstrate that the performance of NASH is robust to the choice of the only hyperparameter $\lambda$ in App. D.6. Therefore, NASH can be paired with any efficient approximation of Shapley values/semivalues to select data effectively.

## 4 Experiments

In this section, we demonstrate that data selection with NASH leads to substantial improvements over selection with Data Shapley and other baselines via top-$m$ heuristic. In the main paper, we provide results on standard data valuation tasks including training logistic regression (**LR**) models on Wind (**WD**) (Vanschoren, Joaquin, 2014d), Pol (**PO**) (Vanschoren, Joaquin, 2014c) and Phoneme (**PM**) (Grin, Leo, 2022) datasets, as well as training ridge regression (**RR**) models on Physiochemical Protein (**PP**) (Fischer, Sebastian, 2022b) and Auction (**AU**) (Fischer, Sebastian, 2022a) datasets. Furthermore, we consider **larger-scale prompt-based finetuning of language models** including BERT (**BT**) (Devlin et al., 2019) and Llama-2-7B (**LM**) (Touvron et al., 2023) models on Rotten Tomatoes Movie Review (**MR**) (Pang & Lee, 2005), Microsoft Research Paraphrase Corpus (**MP**) (Dolan & Brockett, 2005) and Recognizing Textual Entailment (**RT**) (Bentivogli et al., 2009) datasets. **This matches the largest scale of related works on improving the efficiency of Data Shapley to the best of our knowledge.** We use notations like **WD-LR** to indicate the specific dataset-model combinations. For the efficient approximation of Shapley values and semivalues (see Sec. 5), we adopt *(truncated) MC sampling* (Ghorbani & Zou, 2019) for Shapley values, *least squares approximation* (Li & Yu, 2024) for semivalues, and *FreeShap* (Wang et al., 2024a) for language models. Detailed descriptions of our setups and more experiments are included in App. D.

**Baselines.** For the standard data valuation tasks (**WD-LR**, **PO-LR**, **PM-LR**), we compare NASH against random selection and vanilla Data Shapley as in Wang et al. (2024c). For the finetuning tasks (**MR-BT**, **MP-BT**, **RT-LM**), we additionally consider other data valuation-based data selection methods (via top-$m$ heuristic) including influence function (Koh & Liang, 2017), TracIn (Pruthi et al., 2020) and Representer Point (Yeh et al., 2018) as in Wang et al. (2024a). We give a brief introduction of these methods in App. B.5.

### 4.1 General Data Selection

In this section, we focus on the general data selection setting on natural datasets, where there is no injected noise and the data quality is less heterogeneous. Prior works (Wang et al., 2024c) claim that Data Shapley performs poorly and no better than random in this setting, which we also analyze in Sec. 3.1. *Is Data Shapley really not better than random and*

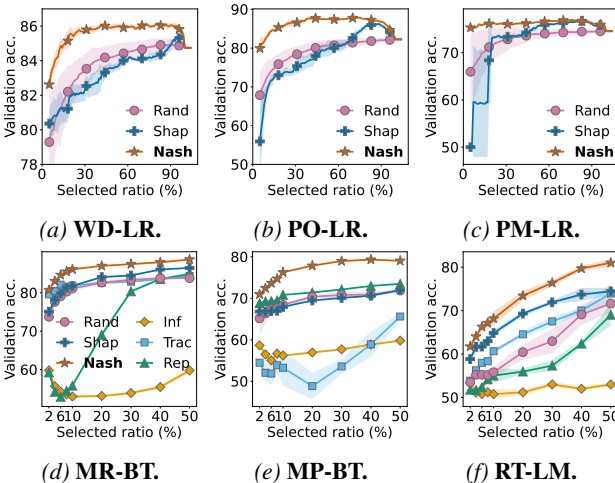

*(a)* **WD-LR.**    *(b)* **PO-LR.**    *(c)* **PM-LR.**

*(d)* **MR-BT.**    *(e)* **MP-BT.**    *(f)* **RT-LM.**

*Figure 5.* **General data selection performance.** Different ratios of training data are selected and evaluated using validation accuracy. NASH consistently outperforms other baselines.

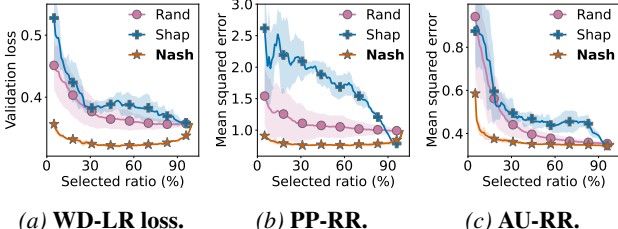

*(a)* **WD-LR loss.**    *(b)* **PP-RR.**    *(c)* **AU-RR.**

*Figure 6.* **Data selection performance using alternative utility functions.** Fig. 6a uses validation loss; Fig. 6b and 6c consider regression tasks and use (negated) mean squared error given by **RR** as the utility function. NASH still consistently outperforms other baselines.

### 4.2 Heterogeneous-Quality Datasets

In Sec. 3.2, we note that prior works (Ghorbani & Zou, 2019; Wang et al., 2024c) demonstrate that Data Shapley works well on heterogeneous-quality datasets with corrupted/noisy data. Since this setting is fairly common for real-life datasets, *is our proposed method* NASH *still more effective than vanilla Data Shapley in this setting?* To answer this question, we inject different proportions of label noises to the training data and perform data selection.

Fig. 7 gives the results. As the proportion of noises increases, the performance of Data Shapley becomes better as expected. Yet, NASH still consistently improves over it. This is because NASH can also identify the corrupted data: These data are likely harmful to all validation data, thus having a small/negative Shapley value w.r.t. every component $u_v$. Adding such data would harm the NASH objective. Therefore, NASH works well with both homogeneous-quality and heterogeneous-quality datasets. Additional results are given in App. D.3.

### 4.3 Compatibility with Other Semivalues

In this section, we investigate whether our NASH framework can be used with other semivalues (e.g., Beta Shapley (Kwon & Zou, 2022), Data Banzhaf (Wang & Jia, 2023)), since all these semivalues (via top-$m$ heuristics) face the same challenge discussed in Sec. 3.1 (i.e., **P1** and **P2**). The results are shown in Fig. 8 (more in App. D.4). Indeed, vanilla semivalues such as Data Banzhaf have ineffective performances due to the issues given in Sec. 3.1 and may perform no better than random. On the other hand, NASH effectively addresses these issues and greatly improves the model performances.

We also include an ablation study in App. D.5.2 to explore how different choices of semivalues may affect data selection performances when paired with NASH. Firstly, we notice that semivalues that consider more interactions and are thus more non-myopic, such as Data Shapley and Data Banzhaf, are better than semivalues that consider less in-

does NASH *solve this issue* (we have argued that NASH is effective in this setting too as it solves **P1** and **P2**)*?* The results are shown in Fig. 5. In all experiments, NASH brings a substantial improvement to all baselines at every selection size, demonstrating its consistent effectiveness in data selection. Additional results are given in App. D.2.

For the larger-scale finetuning tasks, some datasets are inherently of heterogeneous quality to some extent. Thus, vanilla Data Shapley sees a better performance in Fig. 5d and 5f, which confirms its effectiveness on heterogeneous datasets (Wang et al., 2024c). Additionally, it performs better than several other baselines that do not consider interactions (e.g., influence function), which validates its non-myopic and equitable benefits as introduced in Sec. 2. That said, NASH still consistently improves over Data Shapley and outperforms other baselines. This shows that the benefits brought by NASH, such as addressing **P1** and **P2**, are separate from the benefits of Data Shapley. To further validate this, in the following section we further compare the performance of NASH and vanilla Data Shapley in the setting of heterogeneous-quality datasets.

While our theoretical analysis in Sec. 3 focuses on validation accuracy $u_V$/prediction correctness $u_v$, our insights naturally generalize to other tasks and utility functions: Complex utility functions involve multiple roles which cannot be captured by a single Shapley value due to **P1** and **P2** in Sec. 3.1, while NASH considers the interaction among these roles by non-linearly aggregating them. We empirically verify this in Fig. 6 using alternative utility functions. We observe the same trend: Data Shapley can be worse than random, while NASH remains consistently effective. Additional results are shown in Fig. 12.

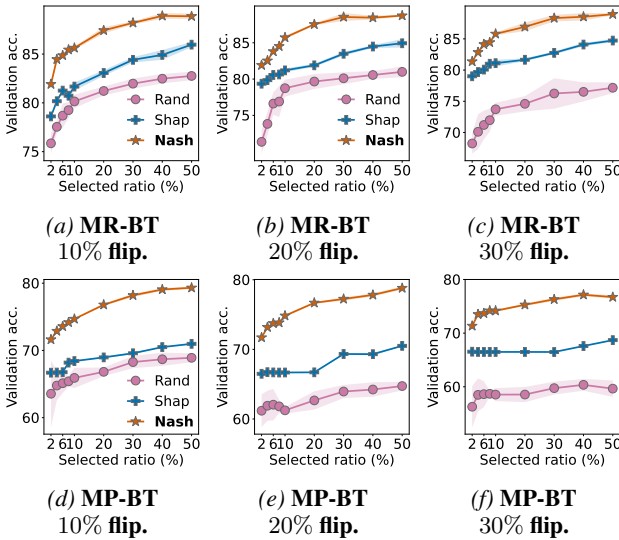

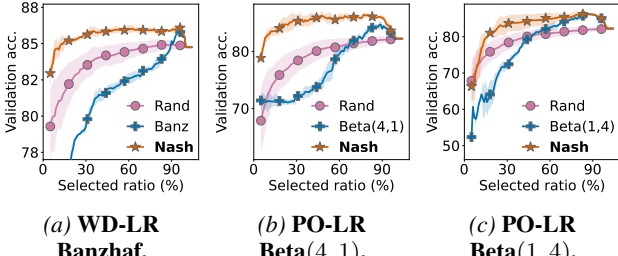

*(a)* **WD-LR Banzhaf.**  *(b)* **PO-LR Beta**(4, 1).  *(c)* **PO-LR Beta**(1, 4).

*Figure 8.* **NASH is compatible with other semivalues.** While other semivalues demonstrate ineffective performances similar to Data Shapley, NASH consistently improves over them.

*(a)* **MR-BT 10% flip.**  *(b)* **MR-BT 20% flip.**  *(c)* **MR-BT 30% flip.**

*(d)* **MP-BT 10% flip.**  *(e)* **MP-BT 20% flip.**  *(f)* **MP-BT 30% flip.**

*Figure 7.* **Data selection performance on MR-BT and MP-BT when label noises are added to the training set.** As the amount of noise increases, NASH consistently outperforms Data Shapley although Data Shapley improves.

teractions such as LOO, even when paired with NASH. Secondly, we discover that when the selection size is very small, semivalues that assign larger weights to smaller coalitions such as Beta$(4, 1)$ could have an advantage. This is because consideration of larger coalitions (thus less myopic) is less useful when only a small number of data are to be selected. However, when the selection size is very large, semivalues that assign larger weights to larger coalitions such as Beta$(1, 4)$ do not lead to good performance. Since these values overlook small coalitions, less helpful data might be chosen at the early stage which negatively affects model performance.

## 5  Related Works

**Efficient approximation of Shapley values.** Exact computation of Shapley values/semivalues is intractable. Existing works on efficient approximation of Shapley values can be branched into two categories, **(1) reducing the times of evaluating** $u_V$ and **(2) using a faster surrogate of** $u_V$. App. B.4 introduces these works in detail. To summarize, works in **(1)** (e.g., Ghorbani & Zou (2019); Li et al. (2026)) aim to approximate Shapley values with guarantees using fewer samples of coalitions $S \subseteq N$ (e.g., $\mathcal{O}(n)$ instead of exponential). Works in **(2)** (Jia et al., 2019a; Wang et al., 2024a) avoid the expensive model retraining (to evaluate $u_V$) by replacing them with faster surrogates. These works complement our work: They can be freely plugged into our work and become more effective in data selection.

**Data selection via top-$m$ heuristic.** Top-$m$ heuristic (Sec. 1) is commonly adopted in data selection beyond Data Shapley and semivalues. For example, *influence*-based methods (Koh & Liang, 2017; Pruthi et al., 2020) are widely used to quantify the impact of individual training data on model performance (which is thus also considered as an approximation to the LOO value); *representer point* (Yeh et al., 2018) decomposes model prediction into a linear combination of training points' representer values. Top-$m$ heuristic then selects data with top-$m$ data values computed using these methods. While App. B.5 introduces these works in detail, they are orthogonal to NASH which is a new data selection framework that replaces top-$m$ heuristic.

## 6  Conclusion and Discussion

In this paper, we analyze why Data Shapley (via top-$m$ heuristic) is not always effective in data selection. We then propose a novel data selection framework NASH which makes Data Shapley and semivalues effective in data selection by decomposing the utility function into simpler Shapley-informative components and aggregating them nonlinearly. Our theoretical and empirical results show the effectiveness of NASH. Future work can explore efficient approximation of Shapley values on different models to complement this work. Future work can also explore whether considering the temporal dependence of each datum's contribution could further improve the efficiency and effectiveness. As NASH is compatible with all semivalues and potentially other data values by aggregating the component values nonlinearly, future work can explore what data values result in the most effective data selection. In App. E, we answer some other questions a reader may have.

## Impact Statement

This paper presents work whose goal is to advance the field of machine learning. There are many potential societal consequences of our work, none of which we feel must be specifically highlighted here.

## Acknowledgments

This research is supported by the National Research Foundation, Singapore under its National Large Language Models Funding Initiative (AISG Award No: AISG-NMLP-2024-001). Xiao Tian and Jue Fan are supported by Agency for Science, Technology and Research (A*STAR) Graduate Academy. The authors would also like to thank the anonymous reviewers and AC for their helpful feedback.

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

# A Overview

## A.1 Summary of Notations

Below gives a summary of notations used in this paper.

---

### Constant & Variables

| | |
|---|---|
| $\alpha$ | Constant in concave functions that fit the learning curve |
| $\beta$ | Constant in concave functions that fit the learning curve |
| $c$ | Curvature of monotone submodular functions |
| $\delta$ | Maximum violation probability allowed for Shapley-informativeness |
| $\epsilon$ | Maximum error allowed for Shapley-informativeness |
| $\mathcal{G}_u$ | Vector that represents the cooperative game induced by utility function $u$ |
| $\gamma$ | Difference between expected sum of Shapley values for subset $\mathcal{M}_1$ and that for subset $\mathcal{M}_0$ |
| $g_i$ | Goodness indicator for datum $i$ |
| $i$ | Datum (to be selected for training) |
| $i^*$ | Datum selected by NASH at each iteration in the algorithm |
| $j$ | Datum (to be selected for training) |
| $\lambda$ | Hyperparameter for NASH |
| $M$ | Subset of data (referring to the candidate selected set) |
| $M^*$ | Optimal subset |
| $\mathcal{M}$ | Randomly sampled subset of data |
| $\mathcal{M}_0$ | Subset whose actual utility is 0 |
| $\mathcal{M}_1$ | Subset whose actual utility is 1 |
| $\mathcal{M}^{\mathrm{I}}$ | Random variable that denotes the subset obtained by independently selecting each train datum with probability $m/n$ |
| $m$ | Selection size |
| $N$ | Feasible set of training data |
| $n$ | Cardinality of $N$; number of feasible data |
| $\phi_u$ | Vector of Shapley values w.r.t. utility function $u$ |
| $\phi_{\max}$ | Maximum absolute $\phi_i(u)$ |
| $S$ | Coalition of data (referring to the coalitions for computing Shapley values/semivalues) |
| $\pi$ | Permutation of training data |
| $\mathcal{S}$ | Randomly sampled coalition |
| $s$ | Size of coalition $S$ |
| $\tau$ | Threshold that informs the sum of Shapley values needed for correct prediction |
| $\tau_v$ | Threshold that informs the sum of Shapley values needed for correct prediction of $v$ |
| $\mathcal{T}$ | Random variable that denotes an unknown threshold $\tau_v$ drawn uniformly at random from $V$ |
| $u_{\max}$ | Maximum utility |
| $V$ | Validation set |
| $\mathcal{V}$ | Variance of $Z$ |
| $v$ | Validation datum |
| $w_s$ | Weighting coefficient to coalitions of size $s$ (for semivalues) |
| $x_i$ | Modular weight of datum $i$ |
| $Z$ | Set of all $Z_i$ for all train datum $i$ |
| $Z_i$ | Random variable that denotes the indicator variable whether $i$ is included/excluded by $\mathcal{M}^{\mathrm{I}}$ when $g_i$ is $+1/-1$. |

---

### Functions & Operators

| | |
|---|---|
| $2^N$ | Power set of $N$ |
| $\Delta_u(\cdot\|\circ)$ | Marginal contribution to coalition $\circ$ brought by datum $\cdot$ w.r.t. utility function $u$ |
| $f(\cdot)$ | Mapping that recovers actual utility from approximated utility |

| | |
|---|---|
| $f_\tau(\cdot)$ | Threshold function that indicates whether $\cdot$ reaches threshold $\tau$ |
| $F_\mathcal{T}(\cdot)$ | Cumulative probability of $\mathcal{T} \leq \cdot$ |
| $\mathbb{I}[\cdot]$ | Indicator variable that returns $1$ if condition $\cdot$ is satisfied and $0$ otherwise |
| $p_\mathcal{T}(\cdot)$ | Probability of $\mathcal{T} = \cdot$ |
| $\pi(\cdot)$ | $\cdot$-th datum in $\pi$ |
| $\pi^{-1}(\cdot)$ | Index of datum $\cdot$ in permutation $\pi$ |
| $\Pr[\cdot]$ | Probability of the event $\cdot$ |
| $\phi_i(u)$ | Shapley value of datum $i$ w.r.t. utility function $u$ |
| $\hat{\phi}_i(u)$ | Estimator of Shapley value of datum $i$ w.r.t. utility function $u$ given by MC sampling |
| $\varphi_i(u)$ | Semivalue of datum $i$ w.r.t. utility function $u$ |
| $S_\cdot^\pi$ | Set of first $\cdot$ data in $\pi$ |
| $u(\cdot)$ | Utility of the set $\cdot$ |
| $\hat{u}(\cdot)$ | Approximated utility of $\cdot$ using sum of Shapley values in $\cdot$ |
| $u_V(\cdot)$ | Validation accuracy on validation set $V$ as utility function |
| $u_v(\cdot)$ | Prediction correctness on validation datum $v$ as utility function |

# B Additional Background and Related Works

## B.1 Equitability Axioms and Other Desirable Properties

The following list of *equitability axioms* ensures that **each datum is valued proportionately to its actual contribution** (Ghorbani & Zou, 2019; Sim et al., 2022; Tian et al., 2024). These equitability axioms originate from the *Shapley fairness axioms* (Shapley, 1953; Dubey et al., 1981). Note that a score being equitable is not a sufficient condition for it to be good for data selection. However, equitability is still desirable to ensure that no datum gains unfair advantages. For example, a low-quality datum should not be selected due to its inflated valuation score.

**[LIN]** *Linearity*: For any two utility functions $u_1$ and $u_2$, $\phi_i(u_1 + u_2) = \phi_i(u_1) + \phi_i(u_2)$.

*Remark.* When two utility metrics are combined without favoring either one, the valuation scores of datum $i$ are also combined without favoring either one. When a utility metric doubles, the valuation score of datum $i$ also doubles.

**[DUM]** *Dummy Player*: If a datum $i$ always contributes the same (i.e., $\Delta_u(i|S) = u(\{i\})$ for all $S \subseteq N \setminus \{i\}$), then its valuation score $\phi_i(u) = u(\{i\})$ (since it is independent of others).

*Remark.* Note that the converse is not true: A datum in general does not *always* make the same contribution (e.g., equal to its valuation score) to every coalition formed by other data.

**[INT]** *Interchangeability*: If data $i$ and $j$ are interchangeable w.r.t. utility function $u$ (i.e., $u(S \cup \{i\}) = u(S \cup \{j\})$ for all $S \subseteq N \setminus \{i, j\}$, then $\phi_i(u) = \phi_j(u)$.

*Remark.* Note that the converse is not true: Two data with the same valuation score are not *always* interchangeable. For example, they could be strong at different roles and one of them would thus be preferred over the other given other selected data.

**[MON]** *Monotonicity*: If $u$ is monotone non-decreasing, then $\phi_i(u) \geq 0$ for any datum $i \in N$.

*Remark.* In this setting, every datum is always making a non-negative contribution and hence has a non-negative valuation score.

Below, we give some other properties that are not generally classified as equitability axioms but are desirable to ensure that each datum's valuation score reflects its true worth.

**[EFF]** *Efficiency*: The sum of valuation scores of all data $i \in N$ is equal to the overall utility $u(N)$, i.e., $\sum_{i \in N} \phi_i(u) = u(N)$.

*Remark.* In the context of ML, this property makes the valuation more interpretable: The overall utility is attributed/broken down to every single datum. However, this property originates from use cases where the overall utility is transferrable (e.g., monetary profits). In ML, the overall utility such as validation accuracy $u_V$ does not need to be transferred and some works (Kwon & Zou, 2022; Wang & Jia, 2023; Tian et al., 2024) deem this property less relevant.

**[DES]** *Desirability* (Carreras & Freixas, 2000): If the marginal contribution of datum $i$ is never smaller than that of datum $j$, then the valuation score of $i$ should not be lower than $j$: $\forall S \subseteq N \setminus \{i, j\} [\Delta_u(i|S) \geq \Delta_u(j|S)] \Rightarrow \phi_i(u) \geq \phi_j(u)$.

*Remark.* Note that the converse is not true: Datum $j$ having a lower score than datum $i$ does not mean $j$ cannot contribute more than $i$ given other selected data. This is actually also a reason why selecting via top-$m$ heuristic might be ineffective.

The Shapley value (Shapley, 1953; Ghorbani & Zou, 2019; Jia et al., 2019a) (Sec. 2) is the only valuation method that jointly satisfies **[LIN]**, **[DUM]**, **[INT]**, **[MON]** and **[EFF]**. It also satisfies **[DES]**. Its relaxation, semivalues (Dubey et al., 1981; Kwon & Zou, 2022; Wang & Jia, 2023) (Sec. 2, App. B.2) is the only valuation method that jointly satisfies **[LIN]**, **[DUM]**, **[INT]** and **[MON]**. It also satisfies **[DES]**.

## B.2 Semivalues

The *semivalues* refer to a family of data values which uniquely satisfy the **[LIN]**, **[DUM]**, **[INT]** and **[MON]** axioms (App. B.1). Similar to the Shapley value, semivalues also consider the marginal contribution of datum $i$ to subsets $S$ of other training data. Formally

**Definition B.1** (Semivalue). The *semivalue* of a datum $i \in N$ with regards to (w.r.t.) utility function $u$ is

$$\varphi_i(u) \coloneqq \sum_{S \subseteq N \setminus \{i\}} w_{|S|} \cdot \Delta_u(i|S)/\binom{n-1}{|S|} , \tag{7}$$

where $w_{|S|}$'s are non-negative *weighting coefficients* such that

$$\sum_{s=0}^{n-1} w_s = 1 . \tag{8}$$

Note that when $|S| = s$ for a fixed $s$, the term $\sum_{S \subseteq N \setminus \{i\}} \Delta_u(i|S)/\binom{n-1}{|S|}$ in Eq. (7) equates to the *average* marginal contribution of datum $i$ to all coalitions of size $s$. Thus, the semivalue is equal to the expected average marginal contribution of datum $i$ when the coalition size $s$ follows a certain probability distribution specified by the weighting coefficients $w_s$'s. For example,

- Data Shapley (Shapley, 1953; Ghorbani & Zou, 2019; Jia et al., 2019b) corresponds to the uniform distribution where all $w_s = 1/n$;
- Data Banzhaf (Wang & Jia, 2023) corresponds to the binomial distribution where $w_s = \binom{n-1}{s}/2^{n-1}$. Alternatively, each coalition $S$ has an equal probability to be drawn;
- Beta Shapley (Kwon & Zou, 2022) corresponds to the beta-binomial distribution controlled by two parameters $\alpha$ and $\beta$. A larger $\alpha$ corresponds to a larger probability on smaller coalition sizes, and vice versa;
- Leave-one-out (LOO) corresponds to the distribution where $w_{n-1} = 1$ and all other $w_s$'s are 0. That is, only the marginal contribution to $N \setminus \{i\}$ is considered.

## B.3 When Selecting via Top-$m$ Heuristic with Data Shapley Works Well

Previous works (Wang et al., 2024c; Chi et al., 2025) present some types of utility functions $u$ where data selection via top-$m$ heuristic with Data Shapley works well. "Works well" means that **(W)** the subset formed by data with top-$m$ Shapley values is optimal or close to optimal. We give a brief summary below.

Wang et al. (2024c) discovers that when $u$ is a *monotonically transformed modular* (MTM) function, the subset formed by data with top-$m$ Shapley values is the optimal size-$m$ subset. Specifically, an MTM function is of the form $u(M) = m(\sum_{i \in M} x_i)$, where $m : \mathbb{R} \to \mathbb{R}$ is monotone and $x_i$ is some fixed modular weight associated with datum $i$. For example, for heterogeneous-quality datasets, the validation accuracy $u_V$ is approximately MTM where the normal data have larger modular weights than the corrupted data.

Chi et al. (2025) identifies that when $u$ is monotone submodular with a smaller *curvature*, the subset formed by data with top-$m$ Shapley values is better (or at least has a better guarantee). Specifically, the curvature $c$ is defined as $c \coloneqq 1 - \min_{i \in N} \frac{\Delta_u(i|N \setminus \{i\})}{\Delta_u(i|\emptyset)}$ (i.e., how much the marginal contribution of $i$, $\Delta_u(i|S)$ would change as $S$ grows larger from the empty set $\emptyset$ to the largest set $N \setminus i$). The smallest $c = 0$ is attained when $u$ is modular (so the denominator is equal to the numerator). In this case, every datum $i$'s Shapley value/semivalue is equal to its modular weights and the subset formed by data with top-$m$ Shapley values is optimal. For general $c \in [0, 1]$, the utility of subset formed by data with top-$m$ Shapley values is at least $(1 - c)^2$ of the utility of the optimal size-$m$ subset.

While these works provide valuable insights about when Data Shapley works well, we stress that **there are two important gaps left unaddressed** by these works which our work successfully closes:

1. The actual utility functions induced by ML (e.g., validation accuracy $u_V$) may not exhibit the above structures. In fact, this is not uncommon as Data Shapley w.r.t. $u_V$ can perform ineffectively in practice. In contrast, our work identifies a property tied to ML-induced utility functions which makes Data Shapley more effective for data selection in ML.
2. To modify vanilla Data Shapley so that it *consistently* works well in data selection, a stricter requirement/property than **(W)** might be desirable. For example, in the context of our analysis, each role/validation datum might prefer different data from the feasible set. Thus, it is impossible for a single selected set to be optimal for every role/validation datum.

In this case, **(W)** becomes less relevant, and we need a stronger property such as the Shapley-informativeness property (Def. 3.1).

Together, closing these two gaps makes our proposed method NASH a substantially more effective Shapley/semivalue-based data selection method.

## B.4 Efficient Approximation of Shapley Values

We recognize existing works on efficient approximation of Shapley values and semivalues as important complements to this work, since these works improve the efficiency of these values, while this work improves the effectiveness of them in data selection. In this section, we give a review of these works, where we focus on those implemented in this work. In particular, we mention in Sec. 5 that they can be branched into two categories as given in the two subsections below.

### B.4.1 REDUCING THE TIMES OF EVALUATING $u$

This line of works focus on reducing the times of evaluating $u_V$ from exponential to polynomial or linear if possible.

**Monte-Carlo (MC) sampling (Shapley, 1953).** An equivalent definition of the Shapley value is

$$\phi_i(u) := \mathbb{E}_\pi[u(S^\pi_{\pi^{-1}(i)}) - u(S^\pi_{\pi^{-1}(i)-1})] = \mathbb{E}_\pi[\Delta_u(i|S^\pi_{\pi^{-1}(i)-1})],$$

where $\pi$ is a permutation of the feasible set drawn uniformly at random, $\pi(k)$ is the $k$-th datum in $\pi$, $\pi^{-1}(i)$ is the index of datum $i$ in permutation $\pi$, $S^\pi_k$ is the set of first $k$ data in $\pi$, i.e., $S^\pi_k = \{\pi(1), \ldots, \pi(k)\}$ and $S^\pi_0 = \emptyset$. Inspired by this definition, MC sampling approximates Shapley values by the average of marginal contributions over $T$ randomly sampled permutations of the feasible set. In each permutation, $u_V$ is evaluated $n$ times where $n$ is the total number of feasible data. Thus, MC sampling efficiently reduces the number of model training from $\mathcal{O}(2^n)$ to $\mathcal{O}(Tn)$. The Shapley value of $i$ is estimated by the average of its marginal contribution in all permutations,

$$\hat{\phi}_i(u) = \frac{1}{T} \sum_{t=1}^{T} \left( u\left(S^{\pi_t}_{\pi_t^{-1}(i)}\right) - u\left(S^{\pi_t}_{\pi_t^{-1}(i)-1}\right) \right).$$

MC sampling yields an unbiased estimate of the Shapley value and the approximation error decreases with a rate of $\mathcal{O}(T^{-1/2})$.

**Truncated MC (TMC) sampling (Ghorbani & Zou, 2019).** TMC sampling leverages the common observation that in ML tasks, when the number of training data is sufficiently high, model performance tends to stabilize. In every permutation, when the marginal contribution of any datum $i$ falls below a threshold, TMC sampling assumes that adding more data into the current coalition will not bring significant utility gains. Thus, all subsequent data in this permutation will be assigned a marginal contribution of $0$ without further model retraining. Since TMC sampling introduces a threshold, this is a biased estimate of Shapley value and the bias is bounded by the threshold.

**Least Squares Approximation (Li & Yu, 2024).** This estimator frames the computation of semivalues as the solution to a least squares regression problem, which aims to minimize the total squared error between the estimated sum of (approximated) semivalues $\sum_{i \in S} \psi_i$ and actual utility $u(S)$ in each subset $S$:

$$\min_{\psi \in \mathbb{R}^n} \sum_{S \subset N; |S| > 0} W_s \left( u(S) - \sum_{i \in S} \psi_i \right)^2,$$

where $W_s$ is a non-negative weight vector associated with coalition size $s = |S|$ and semivalues ($W_s$ can be derived from $w_s$ in Def. B.1) and $\sum_{s=1}^{n-1} W_s > 0$. Instead of sampling permutations, this estimator samples random coalitions directly and accumulates sufficient statistics (i.e., covariance matrix and target vector) required to solve the linear regression problem. The solution to the problem is an unbiased estimator of the semivalues.

### B.4.2 USING A FASTER SURROGATE OF $u$

Due to the prohibitive retraining cost of large ML models, even polynomial times of evaluating $u_V$ is infeasible. Thus, another branch of works focus on estimating $u_V$ without retraining, which greatly improves the computation time.

$K$**NN-Shapley (Jia et al., 2019a).** $K$NN-Shapley is a model-specific algorithm designed for $K$-nearest neighbor ($K$NN) classifiers and can compute the exact Shapley values w.r.t. $K$NN utility function efficiently without model training. It relies on the intuition that a training datum $i$ only impacts the prediction of a validation datum if $i$ is among its $K$ nearest neighbors. The $K$NN utility of a set $S$ on a validation datum $v$ is defined as the proportion of $v$'s $K$ nearest neighbors in $S$ with the same label as $v$, i.e.,

$$u_v(S) = \frac{1}{K} \sum_{k=1}^{\min(K,|S|)} \mathbb{I}[y_{(k)} = y_v], \tag{9}$$

where $y_{(k)}$ is the label of $v$'s $k$-th nearest neighbor. The utility w.r.t. a validation datum $v$ depends solely on the relative ranking of training data's distance to $v$. The algorithm first sorts all training data based on distance to $v$ such that $x_{(1)}$ is the nearest and $x_{(N)}$ is the farthest. The Shapley value of the $i$-th closest datum from $v$ is recursively calculated from the farthest back to the closest,

$$\phi_{x_{(N)}}(u) = \frac{\mathbb{I}[y_{x_{(N)}} = y_v]}{N}, \tag{10}$$

$$\phi_{x_{(k)}}(u) = \phi_{x_{(k+1)}}(u) + \frac{\mathbb{I}[y_{x_{(k)}} = y_v] - \mathbb{I}[y_{x_{(k+1)}} = y_v]}{K} \cdot \frac{\min(K,k)}{k}. \tag{11}$$

By **[LIN]** (App. B.1), Shapley value w.r.t. multiple validation data is the average of the Shapley values w.r.t. each single validation datum. This algorithm is highly efficient because no model retraining/evaluation is required.

**Threshold $K$NN-Shapley (T$K$NN-Shapley) (Wang et al., 2023).** T$K$NN-Shapley is similar to $K$NN-Shapley and is based on threshold $K$NN model (i.e., instead of considering $K$ neighbors, it considers all neighbors whose distance is within a threshold). Likewise, T$K$NN-Shapley can also be calculated exactly and efficiently in an efficient manner.

**FreeShap (Wang et al., 2024a).** FreeShap avoids model retraining entirely and estimates the utility of a subset by approximating neural network finetuning using kernel regression on empirical neural tangent kernels (eNTKs). The marginal contribution of a datum (change in loss/utility) is estimated using kernel regression updates on the eNTK matrix. Since the eNTK of a subset is the submatrix of the full eNTK matrix, marginal contributions can be computed in closed-form linear algebra operations rather than the much more expensive gradient descent in model training. The eNTK is calculated using the Jacobian of the model output w.r.t. the pretrained weights $\theta_0$. For any datum $i$, let the Jacobian be $J(i) := \frac{\partial f(i;\theta_0)}{\partial \theta_0}$. The kernel matrix element between two data $i$ and $j$, $\mathrm{Kernel}(i,j)$, is defined to be the dot product between their Jacobians,

$$\mathrm{Kernel}(i,j) = J(i)J(j)^\top. \tag{12}$$

For a subset of data $S$, the label prediction for a validation datum $v$ is then computed using the eNTK regression model,

$$f_S^{\mathrm{entk}}(v) = \mathrm{Kernel}(v,S)^\top \mathrm{Kernel}(S,S)^{-1} Y_S, \tag{13}$$

where $Y_S$ is the vector of one-hot labels for subset $S$, $\mathrm{Kernel}(S,S)$ is the kernel matrix for subset $S$ where the $i$-th row and $v$-th column is the kernel matrix element between the $i$-th and the $v$-th training data in $S$, and $\mathrm{Kernel}(v,S)$ is a column vector where the $i$-th row is the kernel matrix element between validation datum $v$ and the $i$-th training datum in $S$. The utility of subset $S$ is defined as the negative loss of the predicted label w.r.t. the true label.

FreeShap also leverages TMC sampling to further enhance computational efficiency. As compared to conventional settings in Shapley value-based data selection, FreeShap greatly enhances the scalability of Shapley value-based data selection in terms of dataset size and model size.

### B.5 Other Data Selection Methods via Top-$m$ Heuristic

This section provides a detailed description of other data selection/valuation methods via top-$m$ heuristic which we adopt as baselines in this paper. Unlike Shapley value-based data selection which uses the utility function directly, these methods rely on first-order (gradients), second-order (Hessian) approximations, or feature space to estimate the importance of training data. Under the top-$m$ heuristic, these scores are computed in a one-shot manner and data with the top-$m$ highest scores are selected. These works are orthogonal to NASH which is a new data selection framework that replaces top-$m$ heuristic.

**Influence functions (Koh & Liang, 2017).** Influence functions approximate the effect of upweighting a datum by the change in model parameters without actually retraining. The method uses a second-order Taylor expansion of of the loss function. The influence of a training datum $i$ on the loss of a validation datum $v$ is

$$\text{Inf}(i,v) = -\nabla_\theta L(v, \hat{\theta})^\top \, H_{\hat{\theta}}^{-1} \, \nabla_\theta L(i, \hat{\theta}), \tag{14}$$

where $\nabla_\theta L(v, \hat{\theta})$ is the gradient w.r.t. parameters at the optimal model parameters $\hat{\theta}$, and $H_{\hat{\theta}}^{-1}$ is the inverse Hessian matrix of the training loss. A positive influence score indicates that upweihting datum $i$ would decrease the validation loss, which means datum $i$ is "good".

**TracIn (Pruthi et al., 2020).** TracIn scores estimate influence of a training datum by tracking its contribution to the reduction of validation loss throughout the entire training process. Unlike influence functions that compute all training data's influences once at the end of training, TracIn sums the dot product of gradients at various checkpoints $\theta_t$. The TracIn score of training datum $i$ on validation datum $v$ is given by the sum of first-order approximation for the change in loss $\eta_t \nabla L(i, \theta_t) \, \nabla L(v, \theta_t)$ over all checkpoints $t$:

$$\text{TracIn}(i,v) = \sum_t \eta_t \nabla L(i, \theta_t) \, \nabla L(v, \theta_t), \tag{15}$$

where $\eta_t$ is the learning rate at checkpoint $t$. If the gradient on a training datum aligns with that on a test datum, which yields a positive dot product, then this training datum helps to reduce the loss on the test datum. TracIn scores avoid expensive computation of Hessian matrix and rely solely on first-order gradients computed during the training.

**Representer point selection (Yeh et al., 2018).** Representer point selection decomposes the prediction of neural networks into a linear combination of training data based on the Representer Theorem (Schölkopf et al., 2001). Representer Theorem guarantees that the minimizer of the loss function lies within the subspace spanned by the kernel functions of training data. For a validation datum $v$, the pre-activation output $f(v)$ is

$$f(v) = \sum_{i=1}^{N} \alpha_i \mathbf{f}_i^\top \mathbf{f}_v, \tag{16}$$

where $\alpha_i$ is derived from the loss derivative of training datum $i$ w.r.t. pre-activation, and $\mathbf{f}_i^\top \mathbf{f}_v$ is the similarity kernel between training datum $i$ and validation datum $v$, which is a dot product of their feature embeddings from the second last layer. Then the representer value of any datum $i$ is just its contribution to $f(j)$,

$$\text{Rep}(i) = \alpha_i \mathbf{f}_i^\top \mathbf{f}_v. \tag{17}$$

A high representer score indicates the training datum is similar to the validation datum (i.e., $\mathbf{f}_i^\top \mathbf{f}_v$ is large) or has a high weight $\alpha_i$. This method is highly efficient as it operates on the feature space of the last layer rather than the full parameter space.

# C  Additional Theoretical Results

## C.1  Simple Examples of Shapley-Informative Functions

In Def. 3.1, we give a probabilistic definition of Shapley-informative functions. In this section, we provide simple (but possibly unrealistic) examples to illustrate different levels of Shapley-informativeness. Note that this section is for illustration and understanding of the Shapley-informativeness property only.

As a starting point, a *modular* utility function $u(M) = \sum_{i \in M} x_i$ (where $x_i$ is the modular weight of $i$) is $(0,0)$-Shapley-informative at any size under the identity mapping $f(x) = x$. This is because for such functions, the marginal contribution of any datum $i$ to any coalition $S \subseteq N \setminus \{i\}$ is always equal to $u(\{i\})$, and hence $i$'s Shapley value equals $u(\{i\})$. Then for any $M \subseteq N$, $f(\hat{u}(M)) = \hat{u}(M) = \sum_{i \in M} \phi_i(u) = \sum_{i \in M} u(\{i\}) = u(M)$.

Now consider a near-modular utility function with bounded noise $u(M) = \sum_{i \in M} x_i + E$ such that $E$ is a random noise with $|E| \leq \xi$. The marginal contribution of datum $i$ to any subset $S \subseteq N \setminus \{i\}$ is thus bounded between $x_i \pm 2\xi$. The Shapley value of $i$, $\phi_i(u)$, as an expectation of the marginal contributions, is thus also bounded between $x_i \pm 2\xi$. The sum of Shapley values in a subset $M \subseteq N$, is thus bounded between $\sum_{i \in M} x_i \pm 2|M|\xi$, which differs from $u(M)$ by at most $(2|M|+1)\xi$. Thus, by setting $f$ as the identity mapping $f(x) = x$, $u$ is $((2m+1)\xi, 0)$-Shapley-informative at size $m$ under $f$. When $\xi$ is small, the above property ensures that the sum of Shapley values in $M$ indicates the actual utility of $M$ well. Moreover, if the actual distribution of $E$ is known, a much better $(\epsilon, \delta)$ with $\delta > 0$ is likely to be attainable.

Consider also a dataset which can be partitioned into useless data $N_\times$ and useful data $N_\checkmark$. For every $M \subseteq N_\times$, $u(M) = 0$; whenever $M$ contains at least 1 useful data (i.e., $M \cap N_\checkmark \neq \emptyset$), $u(M) = 100$. Then, every useful datum has a positive Shapley value whereas every useless datum has a 0 Shapley value. By setting $f$ as the mapping $f(x) = 100 \cdot \mathbb{I}[x > 0]$ (where $\mathbb{I}$ is the indicator function), $u$ is $(0,0)$-Shapley-informative at any size.

## C.2  Arbitrary Utility Functions Are Not Shapley-Informative

In Sec. 3.1, we explain why arbitrary utility functions $u : 2^N \to \mathbb{R}$ are in general not Shapley-informative. In this section, we give an example of functions that are not Shapley-informative:

*Table 3.* **A set of games that are not distinguishable from their Shapley value vectors.** $x, y, z \in \mathbb{R}$ are arbitrary numbers.

| Coalition $S$ | $\emptyset$ | $\{1\}$ | $\{2\}$ | $\{3\}$ | $\{1,2\}$ | $\{1,3\}$ | $\{2,3\}$ | $\{1,2,3\}$ |
|---|---|---|---|---|---|---|---|---|
| Utility $u(S)$ | 0 | $-x+z$ | $-x+y$ | 0 | $x$ | $y$ | $z$ | 0 |

Consider Shapley-informativeness for size-2 subsets (i.e., $\{1,2\}, \{2,3\}, \{1,3\}$ and their actual utilities $x, y$ and $z$). For every game of the above form, the Shapley values $\phi_1(u) = \phi_2(u) = \phi_3(u) = 0$. Hence, $\hat{u}(\{1,2\}) = \hat{u}(\{1,3\}) = \hat{u}(\{2,3\}) = 0$. Under any mapping $f$, $f(\hat{u}(\{1,2\})) = f(\hat{u}(\{1,3\})) = f(\hat{u}(\{2,3\})) = f(0)$. Nothing about $x, y$ and $z$ can be informed. Thus, an arbitrary utility function $u$ is not Shapley-informative under any mapping.

*Remark.* Wang et al. (2024c) gives a simpler example of 2 different utility functions with the same Shapley value vectors. Our above example generalizes it to infinite number of utility functions with the same Shapley value vectors.

That said, in Sec. 3.1 we give our view that ML-induced utility functions cannot take arbitrary value. Certain structures of them could make them Shapley-informative under carefully chosen mappings. This view is validated by our theoretical analysis and the empirical effectiveness of our proposed methods.

## C.3  Further Discussion on Roles

In Sec. 3.1, we motivate that commonly used utility functions such as validation accuracy $u_V$ are less likely to be Shapley-informative because it involves multiple roles and each feasible datum $i \in N$ has its unique strength in a certain set of roles. In this section we provide more discussions on these roles.

Conceptually, roles are latent directions in how a training datum can affect the objective $u$. For complex utility functions such as $u_V$, it is evident that multiple roles exist. This can be seen from how each training datum affects the prediction correctness on different validation data. For example, when training datum $i$ is closer to validation datum $v_i$ and training

datum $j$ is closer to validation datum $v_j$, $i$ and $j$ would contribute to $u_V$ differently (e.g., if $v_i$ is already learnt well, then $j$ is likely to have a larger contribution). For simplicity one may think of roles as being orthogonal to each other. Each training datum is then associated with a vector whose $k$-th entry represents its contribution to the $k$-th role. Based on the values at each entry, each training datum has different strengths and different data contribute more to different roles. Putting **P1** and **P2** in this framework, although different training data are associated with different vectors of roles (capturing their unique strengths), their Shapley values can be the same. Thus, vanilla Data Shapley may select data ineffectively.

On the other hand, if there is only one role, each datum's contribution to the role should be strongly correlated to its Shapley value; if the utility function involves fewer roles, Data Shapley will also be more informative. For example, in Sec. 3.2, by breaking down $u_V$ to prediction correctness $u_v$'s, the utility function becomes provably Shapley-informative. Moreover, the aforementioned problems **P1** and **P2** can also be justified by observing the $u_v$'s of each training datum $i \in N$, as shown in Fig. 9. There is a very low correlation between data $i$ and $j$'s Shapley values on each validation datum (the Pearson correlation is as low as 0.176). This shows that $i$ and $j$ indeed contribute to each role of $u_V$ differently. Yet, they cannot be distinguished by Data Shapley since they have the same score, which explains why Data Shapley could be ineffective[5].

*Remark.* Each validation datum may not correspond to a single role. For example, if two validation data are close to each other, a training datum that contributes much to one of them would also contribute much to the other, but the contributions are not exactly the same. This suggests that the two validation data are associated with highly overlapping, but not exactly identical, roles.

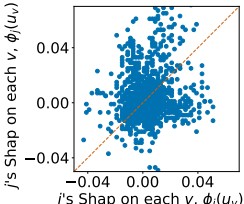

*Figure 9.* **Paired comparison of data $i$ and $j$'s Shapley values $\phi_i(u_v)$ and $\phi_j(u_v)$ on each validation datum $v \in V$.** Specifically, $i$ and $j$ are taken from the **PO-LR** setting, both with Data Shapley values $\phi_i(u_V) = \phi_j(u_V) = 3.6 \times 10^{-3}$. Each scatter point corresponds to a validation datum $v$'s $(\phi_i(u_v), \phi_j(u_v))$. We also plot the line $y = x$ (red dashed line).

## C.4  Empirical Justification of Consistent Player (Assump. 3.2)

Assump. 3.2 assumes that a training datum's contribution to a validation datum is typically consistently good (non-negative marginal contribution) or consistently bad. Empirically, as shown in Tab. 4, the violation frequency is very small, which supports that the assumption is reasonable as an idealized setting for theoretical analysis.

*Table 4.* **Violation frequency of the Consistent Player assumption (WD-LR).** The results are measured on the **WD-LR** dataset (App. D.1.1). We sample 10,000 marginal contributions for each pair of training datum and validation datum.

| Frequency of violations | Min | 25-th percentile | Median | 75-th percentile | Max |
|---|---|---|---|---|---|
| **Across training data** | 1.4e−5 | 1.4e−4 | 2.5e−4 | 3.4e−4 | 1.6e−3 |
| **Across validation data** | 0 | 4.4e−5 | 1.7e−4 | 4.5e−4 | 1.3e−3 |

## C.5  Consistent Player Implies Separation of Conditional Expectation

In this section, we present an important and intuitive lemma that leads to the main Shapley-informativeness proposition (Prop. 3.3). The lemma basically states that the sum of Shapley values of a 0-utility subset $\mathcal{M}_0$, $\hat{u}(\mathcal{M}_0)$, is expected to be smaller than that of a 1-utility subset $\mathcal{M}_1$, $\hat{u}(\mathcal{M}_1)$. Then we can prove Prop. 3.3 by showing that any randomly selected subset $\mathcal{M}$ has its sum of Shapley values $\hat{u}(\mathcal{M})$ concentrated around its conditional expectation in later sections.

To simplify the analysis, we consider each candidate $\mathcal{M}$ is selected by independently deciding whether to include each datum $i \in N$ with probability $m/n$ (we will discuss how to restrict to the constant size-$m$ case later). We use $\mathcal{M}^I$ to denote such a randomly selected subset. A formal statement of our lemma is as follows:

---

[5]In contrast, our method NASH accounts for this: if a validation datum below the $y = x$ line is poorly predicted, then NASH will prefer datum $i$ (which contributes more to this validation datum) over datum $j$.

**Lemma C.1** (Consistent Player Implies Separation of Conditional Expectation). *Let $N := \{1, 2, \cdots, n\}$ be a set of data and $u : 2^N \to \{0, 1\}$ be a utility function mapping each coalition $S \subseteq N$ to its utility $u(S)$. Assume $u$ satisfies the Consistent Player assumption (Assump. 3.2). Let $\mathcal{M}^{\mathrm{I}}$ be the random variable denoting the subset obtained by independently selecting each datum $i \in N$ with probability $m/n$. Then*

$$\mathbb{E}_{\mathcal{M}^{\mathrm{I}}}[\hat{u}(\mathcal{M}^{\mathrm{I}}) \mid u(\mathcal{M}^{\mathrm{I}}) = 0] \le \mathbb{E}_{\mathcal{M}^{\mathrm{I}}}[\hat{u}(\mathcal{M}^{\mathrm{I}}) \mid u(\mathcal{M}^{\mathrm{I}}) = 1],$$

*with equality attained only if all Shapley values are zero ($\forall i \in N \ [\ \phi_i(u) = 0\ ]$).*

*Proof.* Recall that under Assump. 3.2 each player $i \in N$ is associated with a goodness indicator $g_i \in \{+1, -1\}$ characterizing whether it is consistently good or bad. Since the Shapley value is a weighted sum of marginal contributions with non-negative weights, we have the following facts under Assump. 3.2:

$$\begin{aligned}
\forall i \in N \quad & [g_i = +1 \Rightarrow \phi_i(u) \ge 0]; \\
\forall i \in N \quad & [g_i = -1 \Rightarrow \phi_i(u) \le 0].
\end{aligned} \tag{18}$$

That is, the sign of each datum $i$'s Shapley value is the same as its $g_i$.

For each datum $i \in N$ we define the following random variables

$$Z_i = \begin{cases} \mathbb{I}[i \in \mathcal{M}^{\mathrm{I}}] & \text{if } g_i = +1 \\ \mathbb{I}[i \notin \mathcal{M}^{\mathrm{I}}] & \text{if } g_i = -1 \end{cases},$$

where $\mathbb{I}$ is the indicator function ($\mathbb{I}[P] = 1$ if predicate $P$ is true and $0$ otherwise). Clearly, each $Z_i$ is independent of one another since each $i$ is included in $\mathcal{M}^{\mathrm{I}}$ independently. Let $Z := (Z_1, Z_2, \cdots, Z_n)$. Then for each $\mathcal{M}^{\mathrm{I}}$ we can rewrite its utility $u(\mathcal{M}^{\mathrm{I}})$ and its sum of Shapley values $\hat{u}(\mathcal{M}^{\mathrm{I}})$ in terms of $Z_i$ as

$$\begin{aligned}
u(\mathcal{M}^{\mathrm{I}}) &\overset{(A)}{=\!=} u(\{i : g_i = 2Z_i - 1\}) = U(Z); \\
\hat{u}(\mathcal{M}^{\mathrm{I}}) &\overset{(B)}{=\!=} \sum_{\substack{i \in N \\ g_i = +1}} \phi_i(u) \cdot Z_i + \sum_{\substack{i \in N \\ g_i = -1}} \phi_i(u) \cdot (1 - Z_i) = \sum_{i \in N} |\phi_i(u)| \cdot Z_i + \sum_{\substack{i \in N \\ g_i = -1}} \phi_i(u) = \hat{U}(Z).
\end{aligned}$$

Specifically, we partition $\mathcal{M}^{\mathrm{I}}$ based on whether $g_i$ is $+1$ or $-1$, such that when $Z_i = 1$, $g_i = 2Z_i - 1 = +1$; when $Z_i = 0$, $g_i = 2Z_i - 1 =, Z_i = -1$. Step (B) is from Fact. (18). Clearly, both $U(Z)$ and $\hat{U}(Z)$ are monotone increasing with regards to every $Z_i$ because whenever a $Z_i$ changes from $0$ to $1$,

- if $g_i = +1$, it means we are including a good datum $i$, which will not decrease the actual utility $U(Z)$ or the sum of Shapley values $\hat{U}(Z)$;
- if $g_i = -1$, it means we are excluding a bad datum $i$, which will not decrease the actual utility $U(Z)$ or the sum of Shapley values $\hat{U}(Z)$ either.

Since both $U(Z)$ and $\hat{U}(Z)$ are monotone increasing functions of independent $Z_i$, by FKG inequality (Grimmett, 2012),

$$\mathrm{Cov}_Z[U(Z), \hat{U}(Z)] \ge 0. \tag{19}$$

That is, $U(Z)$ and $\hat{U}(Z)$ are non-negatively correlated. Rewriting Eq. (19), we have

$$\begin{aligned}
& \mathrm{Cov}_Z[U(Z), \hat{U}(Z)] \\
={} & \mathbb{E}_Z[U(Z) \cdot \hat{U}(Z)] - \mathbb{E}_Z[U(Z)] \cdot \mathbb{E}_Z[\hat{U}(Z)] \\
={} & \Pr_Z[U(Z) = 1] \cdot \mathbb{E}_Z[\hat{U}(Z)|U(Z) = 1] - \Pr_Z[U(Z) = 1] \cdot \mathbb{E}_Z[\hat{U}(Z)] \\
={} & \Pr_Z[U(Z) = 1] \cdot \Big( \mathbb{E}_Z[\hat{U}(Z)|U(Z) = 1] - (\Pr_Z[U(Z) = 1] \cdot \mathbb{E}_Z[\hat{U}(Z)|U(Z) = 1] + \Pr_Z[U(Z) = 0] \cdot \mathbb{E}_Z[\hat{U}(Z)|U(Z) = 0]) \Big) \\
={} & \Pr_Z[U(Z) = 1] \cdot \Pr_Z[U(Z) = 0] \cdot \Big( \mathbb{E}_Z[\hat{U}(Z)|U(Z) = 1] - \mathbb{E}_Z[\hat{U}(Z)|U(Z) = 0] \Big) \\
={} & \Pr_{\mathcal{M}^{\mathrm{I}}}[u(\mathcal{M}^{\mathrm{I}}) = 1] \cdot \Pr_{\mathcal{M}^{\mathrm{I}}}[u(\mathcal{M}^{\mathrm{I}}) = 0] \cdot \big( \mathbb{E}_{\mathcal{M}^{\mathrm{I}}}[\hat{u}(\mathcal{M}^{\mathrm{I}}) \mid u(\mathcal{M}^{\mathrm{I}}) = 1] - \mathbb{E}_{\mathcal{M}^{\mathrm{I}}}[\hat{u}(\mathcal{M}^{\mathrm{I}}) \mid u(\mathcal{M}^{\mathrm{I}}) = 0] \big).
\end{aligned}$$

When $\Pr_{\mathcal{M}^{\mathrm{I}}}[u(\mathcal{M}^{\mathrm{I}}) = 1] = 0$ or $\Pr_{\mathcal{M}^{\mathrm{I}}}[u(\mathcal{M}^{\mathrm{I}}) = 1] = 0$, the problem is trivial as any selection is optimal. In all non-trivial cases, Eq. (19) implies that

$$\mathbb{E}_{\mathcal{M}^{\mathrm{I}}}[\hat{u}(\mathcal{M}^{\mathrm{I}}) \mid u(\mathcal{M}^{\mathrm{I}}) = 1] - \mathbb{E}_{\mathcal{M}^{\mathrm{I}}}[\hat{u}(\mathcal{M}^{\mathrm{I}}) \mid u(\mathcal{M}^{\mathrm{I}}) = 0] \geq 0,$$

with equality attained when $\mathrm{Cov}_Z[U(Z), \hat{U}(Z)] = 0$. Note that

$$\mathrm{Cov}_Z[U(Z), \hat{U}(Z)]$$

$$= \mathrm{Cov}_Z \left[ U(Z), \sum_{i \in N} |\phi_i(u)| \cdot Z_i + \sum_{\substack{i \in N \\ g_i = -1}} \phi_i(u) \right]$$

$$= \sum_{i \in N} |\phi_i(u)| \cdot \mathrm{Cov}_Z[U(Z), Z_i]$$

$$= \sum_{i \in N} |\phi_i(u)| \cdot (\mathbb{E}_Z[U(Z) \cdot Z_i] - \mathbb{E}_Z[U(Z)] \cdot \mathbb{E}_Z[Z_i])$$

$$= \sum_{i \in N} |\phi_i(u)| \cdot \left( \Pr_Z[Z_i = 1] \cdot \mathbb{E}_Z[U(Z)|Z_i = 1] - \Pr_Z[Z_i = 1] \cdot \mathbb{E}_Z[U(Z)] \right)$$

$$= \sum_{i \in N} |\phi_i(u)| \cdot \Pr_Z[Z_i = 1] \cdot \left( \mathbb{E}_Z[U(Z)|Z_i = 1] - \Pr_Z[Z_i = 1] \cdot \mathbb{E}_Z[U(Z)|Z_i = 1] - \Pr_Z[Z_i = 0] \cdot \mathbb{E}_Z[U(Z)|Z_i = 0] \right)$$

$$= \sum_{i \in N} |\phi_i(u)| \cdot \Pr_Z[Z_i = 1] \cdot \Pr_Z[Z_i = 0] \cdot (\mathbb{E}_Z[U(Z)|Z_i = 1] - \mathbb{E}_Z[U(Z)|Z_i = 0]).$$

The first term $|\phi_i(u)| \geq 0$ and attains $0$ only if $\phi_i(u) = 0$. The second and third terms, $\Pr_Z[Z_i = 1] \cdot \Pr_Z[Z_i = 0] = (m/n) \cdot (1 - m/n) > 0$. Since $U(Z)$ is monotone increasing, the last term always $\geq 0$, where $0$ is attained only if changing $Z_i$ from $0$ to $1$ never increases $U$'s value, regardless of whether other $j \in N \setminus \{i\}$ are selected or not. This means $i$'s marginal contribution is always $0$, hence its Shapley value $\phi_i(u)$ is $0$. Thus, $\mathrm{Cov}_Z[U(Z), \hat{U}(Z)] = 0$ only if every player $i$'s Shapley value is $0$ (which is trivial in our setting). This concludes our proof. □

### C.6 Proof and Discussion of Proposition 3.3

With Lem. C.1, we are now ready to prove that any utility function that satisfies the Consistent Player assumption (Assump. 3.2) is Shapley-informative. The idea is to show that for any candidate set $M \subseteq N$, its sum of Shapley values $\hat{u}(M)$ concentrates around its conditional expectation $\mathbb{E}_{\mathcal{M}}[\hat{u}(\mathcal{M}) \mid u(\mathcal{M}) = u(M)]$.

**Proposition 3.3** (Consistent Player Implies Shapley-Informativeness, Formal). *Let $N := \{1, 2, \cdots, n\}$ be a set of data and $u : 2^N \to \{0, 1\}$ be a utility function mapping each coalition $S \subseteq N$ to its utility $u(S)$. Assume $u$ satisfies the Consistent Player assumption (Assump. 3.2). For $\tau \in \mathbb{R}$, let $h_\tau$ be the threshold function $h_\tau(u(M)) = \mathbb{I}[u(M) \geq \tau]$. For any selection size $m \leq n$, there exists some threshold $\tau$ such that $u$ is $(0, \delta)$-Shapley-informative at size $m$ under mapping $h_\tau$, where*

$$\delta = \exp\left(-\frac{\gamma^2}{8\mathcal{V} + \frac{4}{3}\phi_{\max}\gamma}\right);$$

$$\gamma := \mathbb{E}_{\mathcal{M}^{\mathrm{I}}}[\hat{u}(\mathcal{M}^{\mathrm{I}}) \mid u(\mathcal{M}^{\mathrm{I}}) = 1] - \mathbb{E}_{\mathcal{M}^{\mathrm{I}}}[\hat{u}(\mathcal{M}^{\mathrm{I}}) \mid u(\mathcal{M}^{\mathrm{I}}) = 0];$$

$$\mathcal{V} := \frac{m(n - m)}{n^2} \sum_{i \in N} |\phi_i(u)|^2;$$

$$\phi_{\max} := \max_{i \in N} |\phi_i(u)|.$$

*Proof.* By Lem. C.1, in the non-trivial cases, $\gamma > 0$. Choose $\tau = \mathbb{E}_{\mathcal{M}^{\mathrm{I}}}[\hat{u}(\mathcal{M}^{\mathrm{I}}) \mid u(\mathcal{M}^{\mathrm{I}}) = 0] + \gamma/2$. From App. C.5, $\hat{u}(\mathcal{M}^{\mathrm{I}}) = \hat{U}(Z) = \sum_{i \in N} |\phi_i(u)| \cdot Z_i + \sum_{\substack{i \in N \\ g_i = -1}} \phi_i(u)$. Treating each $|\phi_i(u)| \cdot Z_i$ as an independent random variable bounded by $\phi_{\max}$. The variance of $|\phi_i(u)| \cdot Z_i$ is thus $(m/n) \cdot (1 - m/n) \cdot |\phi_i(u)|^2$. By Bernstein's inequality (Bernstein,

1924) (one-sided version),

$$\Pr_{\mathcal{M}^{\mathrm{I}}}[\hat{u}(\mathcal{M}^{\mathrm{I}}) \geq \tau \mid u(\mathcal{M}^{\mathrm{I}}) = 0]$$

$$= \Pr_{Z}[\hat{U}(Z) - \mathbb{E}_Z[\hat{U}(Z) \mid U(Z) = 0] \geq \frac{\gamma}{2} \mid U(Z) = 0]$$

$$\leq \exp\left(-\frac{(\frac{\gamma}{2})^2}{2\sum_{i \in N} \frac{m}{n} \cdot (1 - \frac{m}{n}) \cdot |\phi_i(u)|^2 + \frac{1}{3}\phi_{\max}\gamma}\right)$$

$$\leq \exp\left(-\frac{\gamma^2}{8\mathcal{V} + \frac{4}{3}\phi_{\max}\gamma}\right).$$

Similarly,

$$\Pr_{\mathcal{M}^{\mathrm{I}}}[\hat{u}(\mathcal{M}^{\mathrm{I}}) \leq \tau \mid u(\mathcal{M}^{\mathrm{I}}) = 1] \leq \exp\left(-\frac{\gamma^2}{8\mathcal{V} + \frac{4}{3}\phi_{\max}\gamma}\right).$$

$\square$

*Remark.* As an example, consider the task of selecting 400 out of 2000 data with $\gamma = 0.25, \mathcal{V} = 1.09e{-}2, \phi_{\max} = 1.05e{-}2$,[6] the bound given in Prop. 3.3 gives $\delta \approx 4.08e{-}3$.

*Remark.* Prop. 3.3 analyzes size-$m$ subsets by consider binomial samples where each datum is included with probability $m/n$ for ease of analysis. While the bound is meaningful, if one wants to strictly enforce a size-$m$ subset $\mathcal{M}$ instead of a binomal sample $\mathcal{M}^{\mathrm{I}}$, a straightforward way is to condition on the event $|\mathcal{M}^{\mathrm{I}}| = m$. For example, by Stirling,

$$\Pr[|\mathcal{M}^{\mathrm{I}}| = m] = \binom{n}{m}\left(\frac{m}{n}\right)^m\left(1 - \frac{m}{n}\right)^{n-m}$$

$$= \frac{n!}{m!(n-m)!}\left(\frac{m}{n}\right)^m\left(1 - \frac{m}{n}\right)^{n-m}$$

$$\geq \frac{\sqrt{2\pi n}n^n e^{-n}e^{\frac{1}{12n+1}}}{\sqrt{2\pi m}m^m e^{-m}e^{\frac{1}{12m}}\sqrt{2\pi(n-m)}(n-m)^{n-m}e^{m-n}e^{\frac{1}{12(n-m)}}}\left(\frac{m}{n}\right)^m\left(1 - \frac{m}{n}\right)^{n-m}$$

$$\geq \frac{\exp\left(\frac{1}{12n+1} - \frac{1}{12m} - \frac{1}{12(n-m)}\right)}{\sqrt{2\pi m(n-m)/n}}.$$

The last two inequalities are due to Stirling's formula (Robbins, 1955). Hence,

$$\Pr_{\mathcal{M}}[\hat{u}(\mathcal{M}) \geq \tau \mid u(\mathcal{M}) = 0]$$

$$\leq \frac{\Pr_{\mathcal{M}^{\mathrm{I}}}[\hat{u}(\mathcal{M}^{\mathrm{I}}) \geq \tau \mid u(\mathcal{M}^{\mathrm{I}}) = 0]}{\Pr_{\mathcal{M}^{\mathrm{I}}}[|\mathcal{M}^{\mathrm{I}}| = m]}$$

$$\leq \frac{\exp\left(-\frac{\gamma^2}{8\mathcal{V} + \frac{4}{3}\phi_{\max}\gamma}\right) \cdot \sqrt{2\pi m(n-m)/n}}{\exp\left(\frac{1}{12n+1} - \frac{1}{12m} - \frac{1}{12(n-m)}\right)} \tag{20}$$

$$= \frac{\exp\left(-\frac{\gamma^2}{8\mathcal{V} + \frac{4}{3}\phi_{\max}\gamma} + \frac{1}{2}\log(2\pi m(n-m)/n)\right)}{\exp\left(\frac{1}{12n+1} - \frac{1}{12m} - \frac{1}{12(n-m)}\right)}. \tag{21}$$

The same holds for the $u(\mathcal{M}) = 1$ case, *mutatis mutandis*. Note that the above is a worst-case bound where the behavior on size-$m$ could be entirely different from the Bernoulli samples which is unrealistic. Despite so, the bound itself could be practical: For example, consider the task of selecting 400 out of 2000 data with $\gamma = 0.25, \mathcal{V} = 5e{-}3, \phi_{\max} = 3e{-}3$,[7] the bound given in Prop. 3.3 gives $\delta \approx 1.2e{-}4$ while the above bound (21) gives $\delta \approx 6e{-}3$.

---

[6]For $n = 2000$, by Efficiency axiom of the Shapley value (App. B.1), the average Shapley value of all feasible data is $1/2000 = 5e{-}4$. So we consider samples where the Shapley values are drawn from the uniform distribution $\mathrm{Unif}(-9.5e{-}3, 1.05e{-}2)$ and take the sampled value of $\mathcal{V}$ and $\phi_{\max}$.

[7]For $n = 2000$, by Efficiency axiom of the Shapley value (App. B.1), the average Shapley value of all feasible data is $1/2000 = 5e{-}4$. So we consider samples where the Shapley values are drawn from the uniform distribution $\mathrm{Unif}(-2e{-}3, 3e{-}3)$ and take the sampled value of $V$ and $\phi_{\max}$.

*Remark. How do the above bounds change as $n$ scales?* Due to the Efficiency axiom of the Shapley value (App. B.1), the sum of all $\phi_i(u)$'s is constant (1 for a validation datum that is eventually predicted correctly). Thus, in the ML setting where each datum has a limited influence, both $\mathcal{V}$ and $\phi_{\max}$ are of order $\mathcal{O}(n^{-1})$. Given the same selection ratio $m/n$ and $\gamma$, the exponent in the bound in Prop. 3.3 would be of order $\mathcal{O}(n)$ and gets smaller as $n$ grows. For the bound in Eq. (21), the additional term in the exponent is of order $\log n$ (because $m(n-m)/n = (m/n) \cdot ((n-m)/n) \cdot n$ where the first two terms are constant when the selection ratio does not change). It is thus dominated by the first term in the exponent, and overall $\delta$ still decays as $n$ increases.

### C.7  Effectiveness of the Greedy Algorithm

In this section, we provide some supplementary results that demonstrate the effectiveness of the greedy algorithm in optimizing the NASH objective (Def. 3.4). Specifically, when all Shapley values $\phi_i(u_v)$'s are non-negative, the NASH objective is monotone submodular and the greedy algorithm enjoys a guarantee of $(1 - 1/e)$ of the optimum. When some Shapley values are negative for real-life datasets, we plot in Fig. 10 the NASH objective values given by greedily selected subsets vs. 100,000 randomly selected subsets. As can be seen, the greedy algorithm consistently gives a much higher objective value, which validates its efficacy. In fact, this empirical trend appears better than that of many functions that are always monotone submodular. Additionally, the change of NASH objective value as more data are selected is similar to the change of actual utility/validation accuracy.

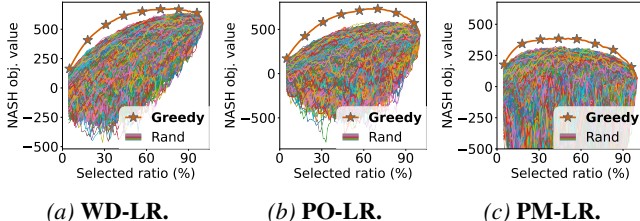

*(a)* **WD-LR.**         *(b)* **PO-LR.**         *(c)* **PM-LR.**

*Figure 10.* **Comparison of the objective value given by greedily selected subsets vs. randomly selected subsets.** The greedy algorithm consistently gives a much higher objective value.

To gain more insights on why the greedy algorithm performs well for the NASH algorithm, we investigate the source of non-submodularity. Clearly, a large number of negative Shapley values come from the noisy/low-quality data with negative/small $\phi_i(u_V)$'s. *Does the* NASH *objective have a nicer structure if such data are removed from the feasible set $N$?* We take the approach of Summers & Kamgarpour (2019) to empirically estimate the submodularity ratio of the NASH objective after removing different proportions of data with bottom Shapley values. Specifically, we estimate $\min_{A \subseteq N, B \subseteq N \setminus A} \left( \sum_{i \in B} [f(A \cup \{i\}) - f(A)] \right) / (f(A \cup B) - f(A))$ by sampling multiple $B$ and $S$'s. The results are shown in Tab. 5. As the removed proportion of data with bottom Shapley values increases, the submodularity ratio increases to close to 1 (i.e., nearly perfectly submodular). Since the removed data are the (consistently) harmful ones and would not be preferred by the greedy algorithm anyway, this can help explain why the greedy algorithm works well.

*Table 5.* **Empirical submodularity ratio as different proportions of low-quality data (indicated by low Shapley values) are removed from the feasible set on WD-LRs.**

| Proportion removed | 0% | 10% | 20% | 30% | 40% | 50% | 60% | 70% | 80% | 90% |
|---|---|---|---|---|---|---|---|---|---|---|
| **Submodularity ratio** | $-6.28$ | 0.63 | 0.85 | 0.88 | 0.91 | 0.93 | 0.95 | 0.97 | 0.98 | 0.99 |

# D    Additional Experiment Results

We run the experiments on two machines (Ubuntu 22.04.3 LTS, AMD EPYC 7763 64-Core Processor with NVIDIA L40 GPUs (CUDA 12.3)). The use of GPUs is required for computing eNTKs for FreeShap (Wang et al., 2024a) and finetuning language models. We write all the programs in Python and execute them using Anaconda. Readers may find the implementation details from `https://github.com/snoidetx/nash`.

## D.1    Setup

### D.1.1    DATASETS

As introduced in Sec. 4, we evaluate our method on standard data valuation datasets as well as larger-scale language datasets. The data valuation datasets are listed in Tab. 6 (they are also included in the OpenDataVal benchmarks (Jiang et al., 2023)). We preprocess each dataset in the same way as existing data valuation works (Kwon & Zou, 2022; Wang & Jia, 2023; Wang et al., 2024c): We pick a size-200 subset as the training set and a size-2000 subset as the validation set. For multi-class datasets, we binarize the labels using $\mathbb{I}[y = 1]$. Moreover, since most if not all existing data valuation works/OpenDataVal do not consider any regression dataset, we include three OpenML datasets of similar scale (the last three rows of Tab. 6).

*Table 6.* **List of standard data valuation datasets used in this paper.**

| Dataset | Acronym | Source |
|---|---|---|
| Wind (Vanschoren, Joaquin, 2014d) | **WD** | https://www.openml.org/d/847 |
| Pol (Vanschoren, Joaquin, 2014c) | **PO** | https://www.openml.org/d/722 |
| Phoneme (Grin, Leo, 2022) | **PM** | https://www.openml.org/d/43973 |
| CPU (Vanschoren, Joaquin, 2014b) | **CP** | https://www.openml.org/d/761 |
| 2D Planes (Vanschoren, Joaquin, 2014a) | **TP** | https://www.openml.org/d/727 |
| APS Failure (Gijsbers, Pieter, 2018) | **AF** | https://www.openml.org/d/41138 |
| Vehicle (Duarte & Hu, 2004) | **VE** | https://www.openml.org/d/357 |
| Credit Card (Dal Pozzolo et al., 2015) | **CC** | https://www.openml.org/d/42397 |
| California Housing (Gazioglu, Mine, 2022) | **CH** | https://www.openml.org/d/43939 |
| Physiochemical Protein (Fischer, Sebastian, 2022b) | **PP** | https://www.openml.org/d/44963 |
| Auction (Fischer, Sebastian, 2022a) | **AU** | https://www.openml.org/d/44958 |

For the language datasets, we use the datasets from Wang et al. (2024a). The Rotten Tomatoes Movie Review (**MR**) dataset contains single-sentence movie reviews labelled as "terrible" or "great". The Microsoft Research Paraphrase Corpus (**MP**) dataset contains sentence pairs such that each pair is associated with a binary label indicating whether the pair of sentences are semantically similar. The Recognizing Textual Entailment (**RT**) dataset also contains sentence pairs and a binary label is used to indicate whether the first sentence entails the second. The Stanford Sentiment Treebank v2 (**ST**) dataset is similar to the **MR** dataset, which contains sentences labelled with their sentiments. Tab. 7 gives the number of feasible data and validation data for each language dataset.

*Table 7.* **List of language datasets used in this paper.**

| Dataset | Acronym | No. of feasible data | No. of val. data |
|---|---|---|---|
| Rotten Tomatoes Movie Review (Pang & Lee, 2005) | **MR** | 8530 | 1066 |
| Microsoft Research Paraphrase Corpus (Dolan & Brockett, 2005) | **MP** | 3668 | 1725 |
| Recognizing Textual Entailment (Bentivogli et al., 2009) | **RT** | 2490 | 277 |
| Stanford Sentiment Treebank v2 (Socher et al., 2013) | **ST** | 25000 | 872 |

### D.1.2 MODELS

For the standard data valuation tasks, we use the logistic regression (**LR**) model as used in (Kwon & Zou, 2022; Wang & Jia, 2023; Wang et al., 2024c). We also considered $K$NN (**KN**), threshold $K$NN (**TN**) and ridge regression (**RR**) models. For the larger-scale finetuning tasks, we follow the same settings as Wang et al. (2024a) and use the BERT (**BT**) (Devlin et al., 2019) and Llama2-7B (**LM**) (Touvron et al., 2023) models. We use `dataset-model` (e.g., **WD-LR**, **MR-BT**) to indicate the specific setting used in each experiment.

For BERT, we finetune the `bert-base-uncased` model. We freeze the first 8 out of the 12 layers of encoders and conduct finetuning on the remaining layers using Adam optimizer. Dropout and data shuffling are disabled. The learning rate is set to be $1e-5$, batch size is 16, number of training epochs is 10. For Llama2-7B, we do not freeze the 32 layers of decoders since we use LoRA for efficient fine-tuning. We set the rank and alpha value to be 16 for LoRa and set no bias and dropout. The learning rate is set to be $1e-5$ for **RT** and **MP** and $1e-6$ for **MR**. The batch size is 2 for **RT** and **MP** and 4 for **MR**. Number of training epochs is 5. The maximum sequence length is set to be 64 for **MR** and **ST**, 128 for **MP** and 256 for **RT**. We adopt the same prompts for each dataset in Wang et al. (2024a) as shown in Tab. 8.

*Table 8.* **List of prompts and targets for finetuning tasks on language datasets.** For single-sequence tasks, `[text]` denotes the input text; for sentence-pair tasks, `[text0]` and `[text1]` represent the individual segments within each pair. `mask` indicates the position of the token to be predicted.

| Model | Dataset | Prompt | Target |
|---|---|---|---|
| BERT (**BT**) | **MR** | `[text]` It was `[mask]`. | terrible, great |
| | **MP** | `[text0]` `[mask]`, `[text1]` | No, Yes |
| | **RT** | `[text0]`? `[mask]`, `[text1]` | No, Yes |
| | **ST** | `[text]` It was `[mask]`. | terrible, great |
| Llama2-7B (**LM**) | **MR** | `[text]` It was `mask`. | terrible, great |
| | **MP** | `[text0]` Question: `[text1]` Yes or No? `mask` | No, Yes |
| | **RT** | `[text0]`? `[text1]` Entailment or not? `mask` | No, Yes |

### D.1.3 DATA VALUATION METHODS

We first give the approximation methods for Data Shapley and semivalues used in our experiments. For the standard data valuation tasks, we adopt MC Shapley with 500 sampled permutations for **LR** models and $K$NN-Shapley and T$K$NN-Shapley for the corresponding nearest neighbor models. For the finetuning tasks, we adopt FreeShap paired with TMC Shapley with 1000 sampled permutations for **MR**, **MP** and **RT**, and 200 sampled permutations for **ST**. The tolerance of TMC Shapley is set to be 0.05. For all experiments that involve other semivalues, we adopt the least squares approximation of semivalues.

Hyperparameters for the other baselines are given in Tab. 9. Influence function and TracIn are computed on unfrozen model parameters. For influence function, we use the conjugate gradient approximation for BERT (**BT**) and the LiSSA approximation for Llama2-7B (**LM**).

*Table 9.* **List of hyperparameters for baseline data valuation methods.**

| Model | Method | Hyperparameters |
|---|---|---|
| BERT (**BT**) | Influence | damping $= 3 \times 10^{-3}$, maximum iteration $= 1000$ |
| | TracIn | checkpoint step $= 3$ iterations |
| | Representer | $\lambda = 3 \times 10^{-3}$ for **MP**, $1.56 \times 10^{-3}$ for **MR**, $2.5 \times 10^{-3}$ for **RT**, $1.56 \times 10^{-3}$ for **ST** |
| Llama2-7B (**LM**) | Influence | damping $= 3 \times 10^{-3}$, repeat $= 1$, depth $= 2500$, scale $= 1 \times 10^{-4}$ |
| | TracIn | checkpoint step $= 1$ epoch |
| | Representer | $\lambda = 2.2 \times 10^{-2}$ for **MP**, $2.2 \times 10^{-2}$ for **MR**, $2.26 \times 10^{-2}$ for **RT** |

## D.2    General Data Selection

In Fig. 11, we include additional experiment results under the general data selection setting described in Sec. 4.1. Similar to the results in the main paper, NASH generally outperforms other baselines by a clear margin and consistently improves over the performance of vanilla Data Shapley. This proves the efficacy of our method.

Note that the performance of vanilla Data Shapley vs. random selection is not consistent. This is as expected because even though no noise is injected, different datasets inherently have different quality of data and thus vanilla Data Shapley could sometimes enjoy an advantage.

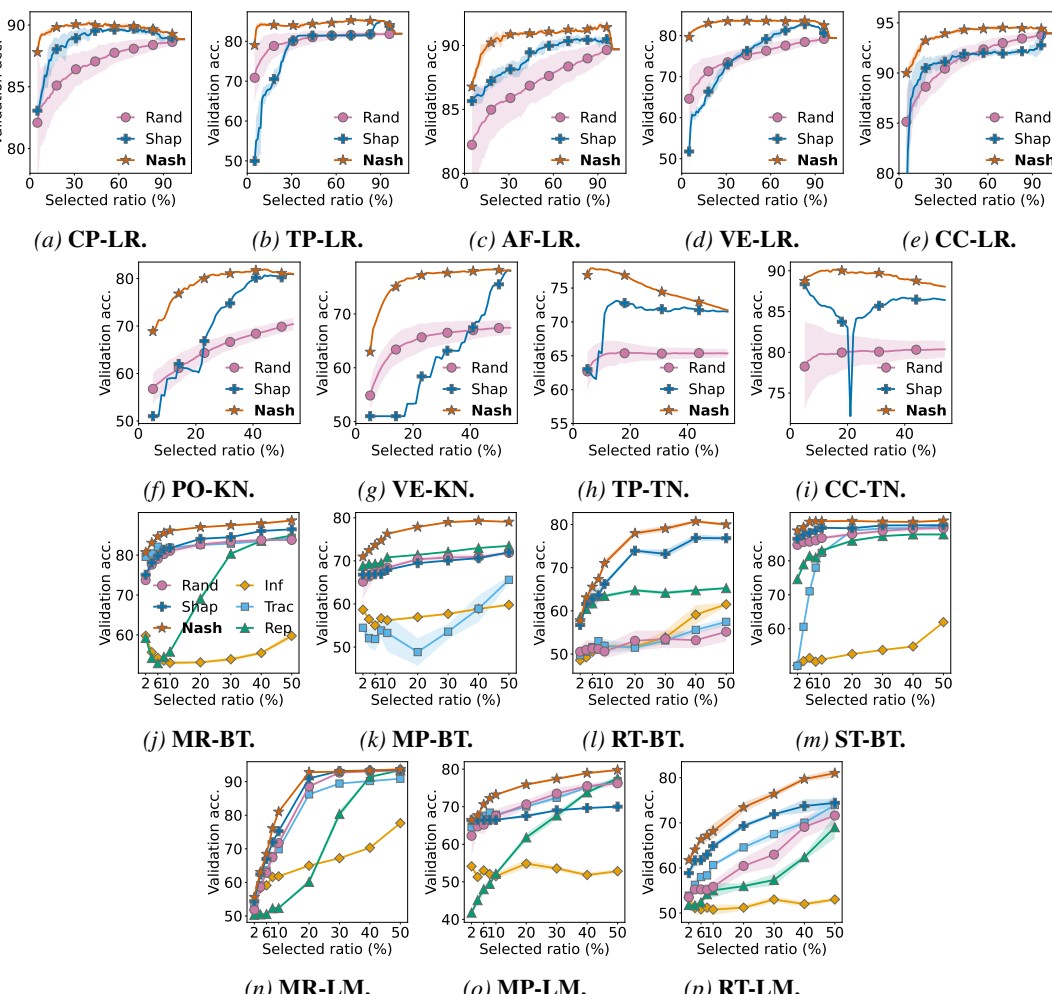

*Figure 11.* **Data selection performance.** Different ratios of training data are selected and evaluated using validation accuracy. NASH in general outperforms other baselines.

Most of our experiments focus on classification problems with validation accuracy as the utility function. Nonetheless, NASH can naturally extend to other problems and is compatible with other utility functions. For example, NASH would prefer data that reduce the mean squared error (MSE) of many poorly predicted validation data instead of well-predicted validation data even though the total change in MSE is the same. To empirically verify this, in Fig. 12 we include additional results for regression tasks with negated MSE as utility function (12a-12c) and for classification tasks but using validation loss as utility function (12d-12e). The trends are similar: Vanilla Data Shapley may work no better than random and NASH always significantly improves over it.

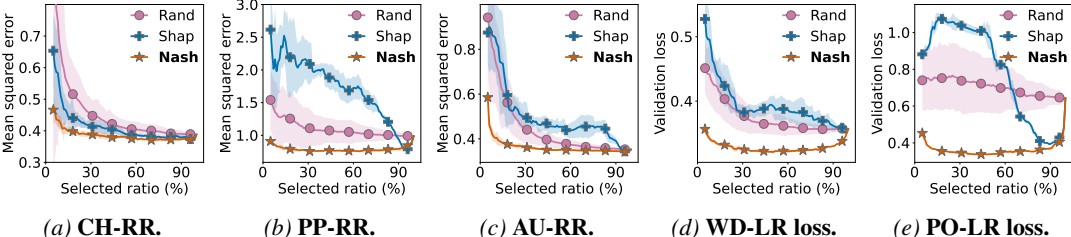

*(a)* **CH-RR.**    *(b)* **PP-RR.**    *(c)* **AU-RR.**    *(d)* **WD-LR loss.**    *(e)* **PO-LR loss.**

*Figure 12.* **NASH improves the performance with other utility functions** $u$. In 12a-12c, we consider regression tasks using the (negated) MSE given by **RR** model as utility function. In 12d and 12e, (negated) loss is used as utility function.

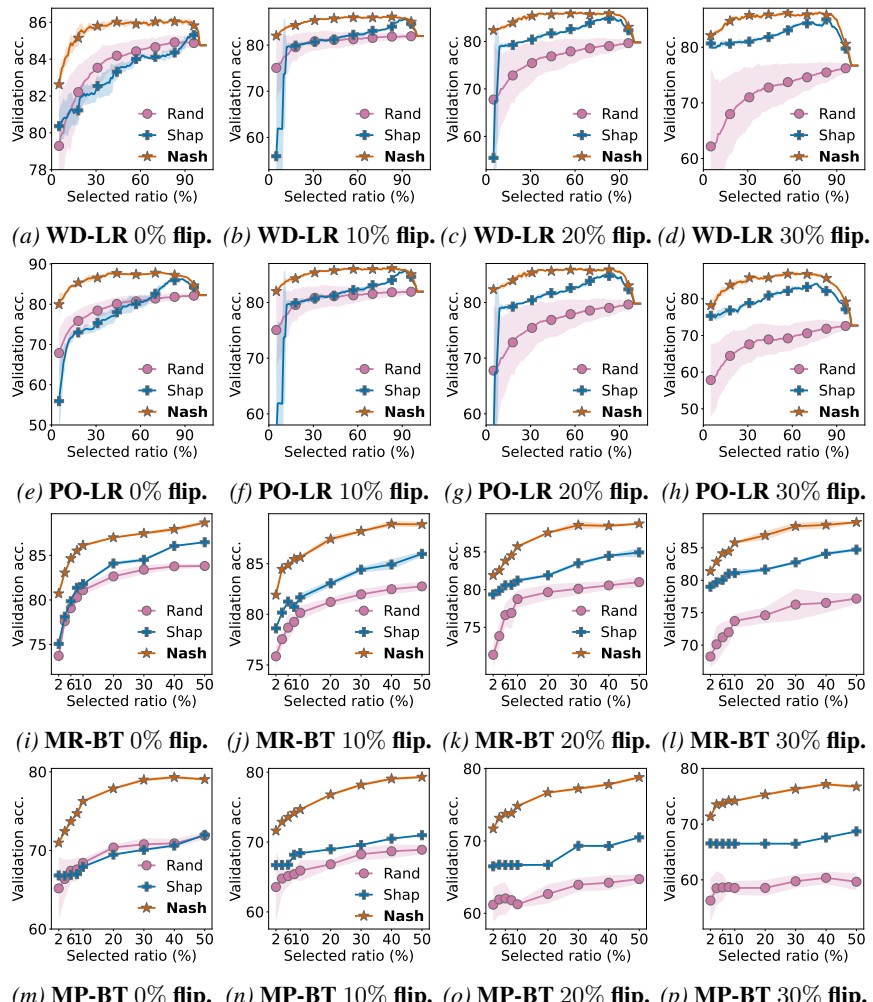

*(a)* **WD-LR** $0\%$ **flip.** *(b)* **WD-LR** $10\%$ **flip.** *(c)* **WD-LR** $20\%$ **flip.** *(d)* **WD-LR** $30\%$ **flip.**

*(e)* **PO-LR** $0\%$ **flip.** *(f)* **PO-LR** $10\%$ **flip.** *(g)* **PO-LR** $20\%$ **flip.** *(h)* **PO-LR** $30\%$ **flip.**

*(i)* **MR-BT** $0\%$ **flip.** *(j)* **MR-BT** $10\%$ **flip.** *(k)* **MR-BT** $20\%$ **flip.** *(l)* **MR-BT** $30\%$ **flip.**

*(m)* **MP-BT** $0\%$ **flip.** *(n)* **MP-BT** $10\%$ **flip.** *(o)* **MP-BT** $20\%$ **flip.** *(p)* **MP-BT** $30\%$ **flip.**

*Figure 13.* **Data selection on heterogeneous datasets**. Different proportions of train labels are flipped to create heterogeneous datasets. Different ratios of training data are selected and evaluated using validation accuracy. Shapley-based data selection performs better than random when datasets are more heterogeneous. NASH consistently improves over Shapley-based data selection.

## D.3 Heterogeneous-Quality Datasets

Data selection experiments on **WD-LR**, **PO-LR**, **MP-BT** and **MR-BT** further validate that Shapley-based data selection may perform no better than random. In this experiment, we flip a portion of the train labels to increase the heterogeneity of datasets. As the ratio of flipped labels increases from $0\%$ to $30\%$ in Fig. 13 and the dataset becomes more heterogeneous, the effectiveness of Shapley value in identifying "good" data becomes more evident in comparison to random selection, which is consistent with Wang et al. (2024c). Even in this setting, NASH consistently improves over the performance of

vanilla Data Shapley, demonstrating its usefulness in improving the effectiveness of Shapley-based data selection on general datasets regardless of the heterogeneity of quality.

### D.4 Compatibility with Other Semivalues

In Sec. 4.3, we mention that since other semivalues also face the issues **P1** and **P2**, NASH can also boost their performances. In this section, we include additional experiment results on more semivalues to validate this. As shown in Fig. 14, NASH can improve a variety of semivalues, including Beta$(16, 1)$ which puts larger weights on smaller coalitions, Banzhaf which puts the same weight on every coalition, and Beta$(1, 16)$ which puts larger weights on larger coalitions.

On the other hand, we also observe that different semivalues have different traits when paired with NASH. For example, the semivalues that put larger weights on larger coalitions, such as Beta$(1, 16)$, still performs ineffectively even after being boosted by NASH. To investigate this and whether Data Shapley has particular advantages, we conduct an ablation study on different choices of semivalues in App. D.5.2, where we discuss the observations in depth.

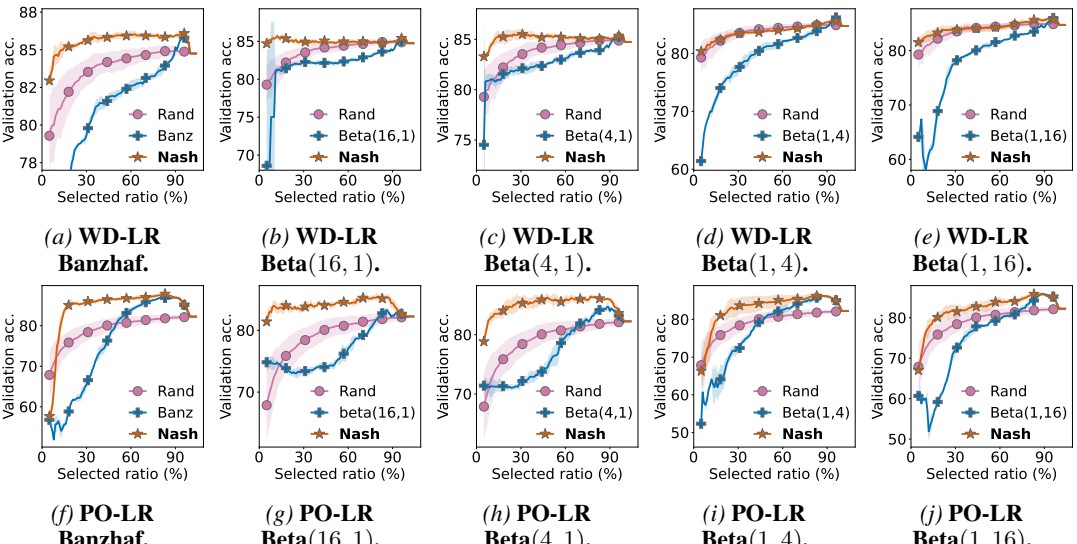

| (a) **WD-LR** Banzhaf. | (b) **WD-LR** Beta$(16, 1)$. | (c) **WD-LR** Beta$(4, 1)$. | (d) **WD-LR** Beta$(1, 4)$. | (e) **WD-LR** Beta$(1, 16)$. |
|---|---|---|---|---|

| (f) **PO-LR** Banzhaf. | (g) **PO-LR** Beta$(16, 1)$. | (h) **PO-LR** Beta$(4, 1)$. | (i) **PO-LR** Beta$(1, 4)$. | (j) **PO-LR** Beta$(1, 16)$. |
|---|---|---|---|---|

*Figure 14.* **NASH improves the performance over other semivalues.**

### D.5 Ablation Studies

#### D.5.1 CHOICE OF $F_\mathcal{T}$ IN THE NASH OBJECTIVE

In this section, we compare across different choices of aggregating function $F_\mathcal{T}$ and justify our choice of the exponential form in the main paper. We consider 3 non-linear functions inspired by the learning curves, including the exponential law (Exp), power law (Pow) and logarithmic law (Log) (Viering & Loog, 2022) (how the learning curve informs about $F_\mathcal{T}$ is described in Sec. 3.3). Specifically, Pow takes the form $F_\mathcal{T}(x) := -x^{-\lambda}$ and Log takes the form $F_\mathcal{T}(x) := -\log x$ (for simplicity we remove the constant and scaling terms which do not impact the selection outcome). We also include the linear aggregation (Lin) (i.e., average), which we argue to be bad in the main paper, for comparison.

Fig. 15 shows the results. The exponential law used in the main paper performs the best. All the non-linear aggregation functions perform better than the linear aggregation, which validates our arguments.

#### D.5.2 CHOICE OF SEMIVALUES $\varphi$

In this section, we investigate the effect of varying the semivalue weights. We focus on how putting larger weights on different coalition sizes would affect data selection performances. As can be seen from Fig. 16, the Shapley value gives the overall best performance. Semivalues that put larger weights on smaller coalitions, such as Beta$(16, 1)$ and Beta$(4, 1)$, tends to have a good performance when the selection ratio is small. This is sensible because the values are significantly dominated by the marginal contributions of data to smaller coalitions, which exactly corresponds to the cases where the

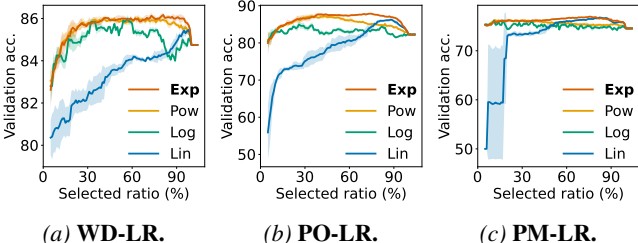

*Figure 15.* **Comparison across different choices of $F_{\mathcal{T}}$.** The exponential form (in the main paper) consistently gives the best result.

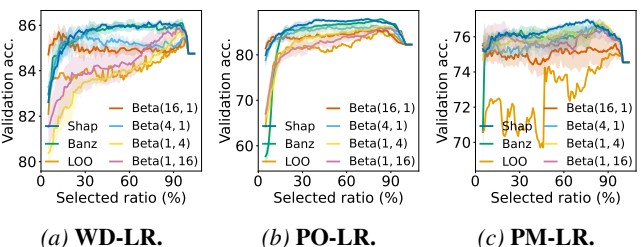

*Figure 16.* **Comparison across different choices of semivalues.**

selection ratio is small. However, as the selection ratio gets larger, these values' performances drop because they fail to consider the long-term interactions among data.

On the other hand, semivalues that put larger weights on larger coalitions, such as Beta$(1, 16)$ and Beta$(1, 4)$, do not in contrast result in a better performance when the selection ratio is large. Instead, their performances tend to be worse for any selection ratio. We hypothesize that this is because of two reasons:

- The contributions of data to larger coalitions can be very different from that to smaller coalitions. Thus, data that contribute more to larger coalitions could be very bad at the early stage, and thus model performance stays low. Even when selection ratio increases, these data may not work well with each other.
- When the coalition size is large, model utility is less affected by a single datum and the marginal contributions may have larger variance. Focusing on such coalitions would make the value more noisy and less useful.

That said, it is still meaningful for future works to explore what semivalues or data values are desirable in data selection.

It is also worth mentioning that by comparing the ablation studies in App. D.5.1 and App. D.5.2, both components of our method NASH, the use of Shapley values and the use of non-linear aggregation, have significant contributions to the selection performance.

### D.6 Sensitivity to Hyperparameter $\lambda$

In this section, we include the performances when different $\lambda$'s in the NASH algorithm (Alg. 1) are used. As can be seen from Fig. 17, NASH's performance is consistently better than vanilla Data Shapley and random selection, across a wide range of $\lambda$'s tested. This shows the practical applicability of our method.

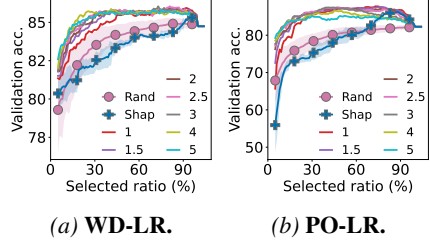

*Figure 17.* **NASH works well for a wide range of $\lambda$'s (i.e., different numbers in the legends).**

# E   Other Questions

> *One of the main draws of Shapley values is their unique satisfaction of equitability axioms (App. B.1). Does* NASH *preserve these axioms?*

These equitability axioms are important in many use cases (e.g., when rewarding each data owner (Sim et al., 2022; Tian et al., 2024)). Since NASH first computes the Shapley values, they can directly be applied to these use cases.

NASH focuses on how to further use (but not change) the values for effective data selection, where existing works simply select top-$m$ data and are ineffective. Under top-$m$, some equitability axioms become undesirable:

- **[EFF]** Efficiency: Kwon & Zou (2022) shows that removing **[EFF]** and placing more weights on smaller coalitions may lead to better performance;
- **[INT]** Interchangeability: Identical data with same value are selected together. However, repetition may only improve on roles that are already well-predicted instead of validation accuracy.

In contrast, NASH can benefit from these axioms. Identical data may not be selected together precisely because they are interchangeable and have same Shapley values for all roles.

> *Is there any other prior work that uses different selection criteria such as diversity-driven selection?*

Yes, there are prior works that use diversity-driven selection in the general field such as Craig (Mirzasoleiman et al., 2020) and model-specific submodular functions (Wei et al., 2015). However, the focus of our work is on **utility-driven selection** (in particular Shapley-based) instead of diversity-driven. We target top-$m$ (Data Shapley) heuristics as (1) they are compatible with all utility functions (some of which might not favor data diversity, e.g., finetuning for specific downstream tasks), (2) Shapley values are non-myopic and equitable (Sec. 2) and (3) yet their data selection performances are inconsistent.

While NASH could ensure better "coverage" of various roles associated with the utility function, it may not always select more diverse data. This is because the validation set may not be evenly/diversely distributed in the data space (e.g., only certain data are helpful to the specific task) and diverse samples may be harmful to the utility function.

> *Would the non-linear aggregation make* NASH *less interpretable then the top-$m$ methods?*

NASH is interpretable since the value of each training datum on each role/validation datum captures their unique strengths. This explains why a datum is preferred over another datum of similar overall score but different strengths by showing what roles the current subset is lacking and needs.

