# OpenReview forum: "Is Data Shapley Not Better than Random in Data Selection? Ask NASH"
_ICML.cc/2026/Conference — ICML 2026 spotlight_

### Official Review · Reviewer_9CMn · 2026-03-02

**Soundness:** 3
**Presentation:** 3
**Significance:** 3
**Originality:** 3
**Overall Recommendation:** 5
**Confidence:** 5

**Summary:**

The paper investigates a critical paradox in data-centric AI: while Data Shapley (DS) is theoretically elegant for data valuation, it frequently underperforms—sometimes failing to beat random selection—in practical data pruning and selection tasks. The authors propose NASH. The contribution is twofold:
It identifies Shapley-informative settings, characterizing the conditions under which DS is expected to provide useful signals. It introduces a novel non-linear aggregation mechanism for Shapley values, moving away from simple top-ranking to a more robust selection strategy that accounts for the non-linear interaction between data points in a training set.

**Compliance With Llm Reviewing Policy:**

Affirmed.

**Key Questions For Authors:**

In the research on Rough Sets, we often distinguish between informative noise and harmful outliers. How does NASH handle data points that have high Shapley values but are actually outliers that might lead to overfitting?
Can NASH be applied to values generated by fast approximation methods like Monte Carlo or Permutation-based samplers? Specifically, does the non-linear aggregation amplify or mitigate the sampling noise?
Could you provide a clearer ablation study showing exactly what the non-linear component captures that a standard weighted ranking misses?
Given the massive scale of modern pre-training, is NASH feasible for selecting subsets of billions of tokens, or is it currently limited to smaller-scale supervised learning?

**Limitations:**

NASH is an aggregation framework; if the underlying Shapley values are fundamentally flawed, NASH may still fail to recover a high-quality subset.
Like many data selection papers, this assumes a static pool of data. In Incremental Learning scenarios, the informativeness of a data point changes as the model evolves. NASH does not yet account for this temporal dependency.
The non-linear aggregation process, while effective, may reduce the transparency of the valuation process, making it harder for humans to interpret why a specific subset was chosen compared to a simple ranking.

**Strengths And Weaknesses:**

The paper addresses a pain point that many practitioners have encountered: the high computational cost of DS often does not translate to superior data selection performance. The shift from linear ranking to non-linear aggregation is a sophisticated step forward. It acknowledges that the value of a data point is not intrinsic but dependent on the neighborhood of other selected points. The definition of Shapley-informative settings provides a much-needed theoretical boundary for when game-theoretic valuation is appropriate. The experiments appear to cover a variety of noise levels and dataset sizes, demonstrating where the random selection baseline fails and where NASH succeeds.
While NASH improves the quality of selection, the underlying calculation of Shapley values remains the primary bottleneck. The paper would benefit from a clearer discussion on whether NASH can work with highly approximated Shapley values. One of the main draws of Shapley values is their unique satisfaction of fairness axioms. Does the non-linear aggregation in NASH preserve these axioms, or is it purely a performance-driven heuristic? The non-linear aggregation layer introduces new hyperparameters. It is unclear how sensitive the system is to these parameters across different data distributions.

---

> ### Author Rebuttal · Authors · 2026-03-31
>
> Thank you for acknowledging the sophistication and significance of our theoretical framework and the comprehensiveness of our experiments! We are glad that you have a very positive opinion about our work.
>
> > Whether NASH can work with highly approximated Shapley values
>
> Yes, NASH is compatible with all efficiently approximated Shapley values/semivalues (Sec. 1). These works can be freely plugged into NASH to be much more **effective**. In Sec. 4 and App. B.4, we describe we use MC sampling for Shapley values, least squares approximation for semivalues, and FreeShap for language models. For example, in Fig. 4a, the error bars of Shapley and NASH come from different MC samples from different seeds, and NASH achieves better performance with smaller standard deviation, which shows NASH's effectiveness is robust against sampling noise.
>
> > Does the non-linear aggregation in NASH preserve the fairness axioms
>
> We agree that the fairness axioms are important (e.g., when rewarding each data owner). Since NASH first computes the Shapley values,they can directly be applied to these use cases.
>
> NASH focuses on how to **further use** (but not change) the values for *effective data selection*, where existing works simply select top-$m$ data and are ineffective. Under top-$m$, some fairness axioms become undesirable:
> - Efficiency: [1] show that removing Efficiency and placing more weights on smaller coalitions may lead to better performance.
> - Interchangeability: Identical data with same value are selected together. However, repetition may only improve on roles that are already well-predicted instead of validation accuracy.
>
> In contrast, NASH can benefit from these axioms. Identical data may not be selected together precisely because they are interchangeable and have same Shapley values for all roles.
>
> > Sensitive to parameters
>
> Please refer to Reviewer ik2q W3 where we discuss the parameters.
>
> > How does NASH handle data points...are actually outliers that might lead to overfitting?
>
> NASH decomposes the complex utility function into simpler roles. We believe harmful outliers have a negative contribution to most roles and would have negative Shapley values [2] across them. Thus, NASH would not prefer the harmful outliers. In contrast, informative data would be beneficial when it is added to a large coalition as it can make the model better at underrepresented roles. The use of Shapley value could help identify such datum as it considers the contribution to various sizes of coalitions. The use of non-linear aggregation also help to priortise data which are good for roles that are currently not well-predicted. Thus, NASH could distinguish the two types of data and correctly prefer informative hard samples over harmful outliers.
>
> > What the non-linear component captures
>
> The non-linear component captures the contribution to different roles and focus on validation data/roles that are not covered by existing selected data. We provide some supplementary results in [THIS LINK](https://anonymous.4open.science/r/nash-rebuttal-E3B0/nash-rebuttal.pdf). Fig. 3 shows that two validation data with the same Shapley value could have entirely different contributions to different roles, and hence they should be differentiated. The subset selected by vanilla Data Shapley would unnecessarily accumulate utilities on some roles, as shown in Fig. 2 (top). By non-linearly aggregating different roles, NASH improves on these underperforming roles (Fig. 2 (bottom)) and leads to a much better performance.
>
> > Is NASH feasible for selecting subsets of billions of tokens, or is it currently limited to smaller-scale supervised learning?
>
> Our work focuses on improving the *effectiveness* of Shapley-based data selection and it has **matched the largest scale** (e.g., LLM finetuning) of related works on improving the efficiency of Data Shapley [3] to the best of knowledge. We note that the research in efficiently approximating Shapley values for massive scale learning tasks (e.g., pretraining) is currently limited. As we highlighted in Sec. 1, future work that fill the gap can be freely plugged into NASH to enable larger-scale data selection.
>
>
> > Like many data selection papers, this assumes a static pool of data... NASH does not yet account for this temporal dependency.
>
> We will mention in our revision that like many data selection papers, the incremental learning scenario is left as future work.
>
> > The non-linear aggregation process makes it harder for humans to interpret why a specific subset was chosen
>
> We can show the value of each training datum on each role/validation datum which captures their unique strengths. Through this, we can explain why a datum is preferred over another datum of similar overall score but different strengths by showing what roles the current subset is lacking and needs.
>
> We hope the above responses help!
>
> [1] https://arxiv.org/pdf/2110.14049
>
> [2] https://arxiv.org/pdf/1904.02868
>
> [3] https://arxiv.org/pdf/2406.04606

---

### Official Review · Reviewer_ik2q · 2026-03-09

**Soundness:** 3
**Presentation:** 3
**Significance:** 3
**Originality:** 3
**Overall Recommendation:** 4
**Confidence:** 3

**Summary:**

The paper proposes a new framework, NASH, to select a subset of high-quality data for prediction tasks. Unlike previous methods that rely on top-m selection based on Data Shapley, NASH finds the limitations of the validation accuracy-based utility function. It introduces a new utility measure that performs better than simply averaging the accuracy across the validation set. By employing a non-linear function to represent this new utility, the framework can find m most useful training points more efficiently. Empirical results show that NASH outperforms the top-m baseline in various settings.

**Compliance With Llm Reviewing Policy:**

Affirmed.

**Final Justification:**

The rebuttal addresses my main concerns sufficiently, and I remain on the positive side.

**Key Questions For Authors:**

Please see above

**Limitations:**

Please see above

**Strengths And Weaknesses:**

Strengths:
1. NASH provides a more efficient and accurate method for data selection.
2. The framework is highly compatible and can be utilized alongside various data importance measures.

Weaknesses:
1. Eq 5 relies on a hard indicator function to determine if the selected subset crosses the threshold $\tau_v$. Eq 6 relaxes this by using an exponential function. Can the exponential law-based $F_T$ really address the goal in Eq 5?

2. The greedy algorithm is used in algorithm 1 to find members in M. However, the approximation guarantees of the greedy algorithm hold if the objective function is monotone submodular, which requires all Shapley values to be non-negative. However, the Shapley data may be negative in some cases, which would violate the guarantee. Can greedy algorithm really help to find near-optimal members in M?

3. What values were used for $\lambda$? Is tuning needed for $\lambda$? If so, it may require a lot of computational overhead. Also, how sensitive is the subset $M$ to different values of $\lambda$?

4. While the standard top-$m$ heuristic serves as a natural baseline, is there any other prior work that uses different selection criteria, for example, diversity-driven selection? In addition to the validation accuracy, does the data subset selected by NASH provide other qualitative properties when compared to the top-m baseline?

---

> ### Author Rebuttal · Authors · 2026-03-31
>
> Thank you for recognizing the efficiency, accuracy, and high compatibility of our method! Here are our responses to your questions and concerns.
>
> > 1. Can the exponential law-based $F_T$ really address the goal in Eq 5?
>
> We would like to clarify that on average, Eq. 6 can address the goal of Eq. 5 as it is derived as follows:
> * In Eq. 5, the hard indicator function depends on the threshold $\tau_v$, which is unknown and is computationally infeasible to estimate;
> * When a variable is unknown, a common practice is to model them as a *random variable* and maximize the expected objective instead. So, we model $\tau_v$ by the random variable $\mathcal{T}$ which denotes the threshold value for any validation point $v$;
> * Instead of maximizing the sum of indicators, we maximize the sum of the expected value of them w.r.t. $\mathcal{T}$.  The expectation of the indicator $\mathbb{I}\left[\hat{u}_v(M) \geq \tau_v\right]$ is $\Pr[\hat{u}_v(M) \geq \mathcal{T}]$;
> * Since $\mathcal{T}$ is a random variable, $\Pr[\hat{u}_v(M) \geq \mathcal{T}]$ corresponds to the c.d.f. $F_\mathcal{T}(\hat{u}_v(M))$.
>
> One key question is how to choose the distribution of $\mathcal{T}$ (or the c.d.f. $F_\mathcal{T}$) well. As we describe in Sec. 3.3 (Line 316 onwards), $F_\mathcal{T}$ can be informed by the learning curves (which has been found to exhibit consistent trend) and should be concave, i.e., more thresholds are likely to be smaller. We compare our choice of an exponential law-based $F_\mathcal{T}$ against alternative forms in App. D.5.1 and it achieves the best overall performance.
>
> > 2. Effectiveness of greedy
>
> The reviewer is right that the approximation guarantees of the greedy algorithm hold if the objective function is monotone submodular and the Shapley data may be negative in some cases. In practice, our experiments show that the *greedy algorithm works well enough empirically* but it is hard to investigate if the members are "near-optimal" as it is computationally infeasible to get the optimal solution for comparison.
>
> We refer the reviewer to App. C.6. where we provide supplementary results that demonstrate the effectiveness of greedy algorithm in optimizing the NASH objective. Fig. 7 shows greedily optimizing NASH objective outperforms 100k randomly selected subsets by a clear margin, which validates its effectiveness although NASH objective may not be strictly monotone submodular. We also analyse that non-submodularity may stem from noisy/low quality data with negative/low Shapley values. As shown in Tab. 4, removing higher proportions of low-quality data leads to higher submodularity ratio, which explains why greedy is effective.
>
> > 3. Values/tuning of $\lambda$
>
> We find that NASH's performance is robust to the choice of $\lambda$ and consistently outperforms vanilla Data Shapley in Fig. 13. Thus, in all experiments, we tune $\lambda$ from a very small candidate set $\{1, 2, 3, 4, 5\}$ and NASH achieves consistently better performances than the baselines. When the compute is highly limited, one may directly use $\lambda = 2$ which we found the most effective in general.
>
>
> > 4. Any other prior work that uses different selection criteria...diversity-driven selection?
>
> Yes, there are prior work that use diversity-driven selection in the general field such as Craig [1] and model-specific submodular functions [2]. However, the focus of our work is on **utility-driven selection** (in particular Shapley-based) instead of diversity-driven. We target top-$m$ (Data Shapley) heuristics as (1) they are compatible with all utility functions (some of which might not favor data diversity, e.g., finetuning for specific downstream tasks), (2) Shapley values are non-myopic and equitable (Sec. 2) and (3) yet their data selection performances are inconsistent.
>
> We understand NASH could ensure better "coverage" of various roles associated with the utility function. However, it may not always select more diverse data. This is because the validation set may not be evenly/diversely distributed in the data space (e.g., only certain data are helpful to the specific task) and diverse samples may be harmful to the utility function.
>
> > 4. Other qualitative properties
>
> Thank you for the interesting question! We contrast the subset selected by NASH and Shapley based on the "coverage of various roles" in Fig. 2 in [THIS LINK](https://anonymous.4open.science/r/nash-rebuttal-E3B0/nash-rebuttal.pdf). NASH improves the prediction (and sum of individual Shapley values) on wrongly predicted roles/validation data. Fig. 3 in the same pdf also shows that data with equal Shapley values may contribute very differently to different roles, which NASH accounts for. We are also open to other suggested qualitative study.
>
> We hope the above responses and additional results improve your opinion of our work. Thank you again and please let us know if you have further questions!
>
> [1] https://arxiv.org/pdf/1906.01827
>
> [2] https://proceedings.mlr.press/v37/wei15.pdf

---

> > ### Author Rebuttal · Reviewer_ik2q · 2026-04-02
> >
> > I would like to thank the authors for their responses. The rebuttal addresses my main concerns sufficiently, and I remain on the positive side.

---

> > > ### Author Response · Authors · 2026-04-08
> > >
> > > Dear Reviewer ik2q,
> > >
> > > Thank you for acknowledging our responses and we are happy that they address your concerns. We appreciate your constructive feedback and will include our clarifications in the revised version. We wish you all the best!
> > >
> > > Regards,
> > >
> > > Authors of Submission \#30815

---

### Official Review · Reviewer_QwZ2 · 2026-03-12

**Soundness:** 3
**Presentation:** 4
**Significance:** 3
**Originality:** 3
**Overall Recommendation:** 5
**Confidence:** 3

**Summary:**

This paper studies the problem of data selection using Shapley values and identify situations where it can perform better than random selection of data. The authors coin the term "Shapley informative" to describe utility functions whose sum of Shapley values for a subset of data can explain the original utility function applied to that subset. The author claims that the failure of Data Shapley is because most utility functions are not Shapley informative.
In this paper, the authors propose a new data selection framework called NASH that is based on breaking down the original utility function (ex: Validation accuracy) into simpler Shapley Informative components. In particular, they show that under certain assumptions, one can break down validation accuracy into several smaller components that are Shapley informative.
Instead of linearly aggregating Shapley-informative components, NASH uses a non linear objective which can better capture the interactions between data points. This objective is then optimized greedily to select the most informative subset of data.

**Compliance With Llm Reviewing Policy:**

Affirmed.

**Final Justification:**

Overall, I lean towards accepting this paper.

My initial concerns were regarding the consistent player assumption. The rebuttal has clarified the necessity of it and it is empirically justified. The authors have also included additional experiments in their supplementary section demonstrating their method works better in tasks other than classification.

Given that my main concerns have been resolved, I maintain my positive evaluation of the work.

**Key Questions For Authors:**

1) Is the consistent player assumption in Theorem 3.1 necessary? I don't believe it is very realistic since based on your subset, a datapoint could either cause the validation datapoint to be correctly/incorrectly classified depending on how diverse the subset is and how much it overfits/underfits.

2) How would this extend to other problems than classification? For example, regression?

**Limitations:**

yes

**Strengths And Weaknesses:**

- The paper clearly explains the issue with original data Shapley technique specifically in the setting of classification problems and validation accuracy. The paper also provides some theoretical backing for their reasoning of why smaller components are Shapley informative albeit with some assumptions. The paper has also conducted several experiments to validate their method and the results show that it performs better than vanilla Shapley and random selection in most cases (Even where vanilla Shapley fails and performs as bad as random selection for instance in the homogenous data setting). The paper also checks the compatibility of their method with other semi-values.

- Assumption 3.2 is quite strong I believe. It assumes that adding a datapoint to a training subset $S$ either provides no new information or changes the classification of a given validation datapoint in a consistent way across all subsets $S$. It cannot cause a misclassification in one subset but a correct classification in another. But in reality, depending on the subset, adding a datapoint could either cause could cause the validation datapoint to be correctly and incorrectly classified depending on S say through overfitting/underfitting. Can you add more details on why this assumption is reasonable? What are some example settings?


- The paper mainly focuses on classification problems. It is not clear to me how this would extend to other problems or other utility functions. For instance, how would this work for regression problems where the utility function is not accuracy but something like mean squared error?

- I believe Figure-6 has incorrect labels on the blue line. The semi-values used does not seem to match the labels.

---

> ### Author Rebuttal · Authors · 2026-03-31
>
> Thank you for appreciating the clarity of our work and the strength of our theoretical and empirical findings. Here are our responses to your concerns and questions.
>
>
> > Is Assump. 3.2 (Consistent Player) in Thm. 3.1 necessary? I don't believe it is very realistic since based on your subset, a datapoint could either cause the validation datapoint to be correctly/incorrectly classified depending on how diverse the subset is and how much it overfits/underfits.
>
> The Consistent Player assumption is *necessary* to **establish a theoretical framework** that designs and analyzes Shapley-informative utility functions. This assumption stems from empirical observations (e.g., Fig. 3(b)) that a training datum's contribution to a validation datum is typically consistently good (non-negative marginal contribution) or consistently bad. We agree with you that this assumption may not *always* hold in practice, thus we include the following empirical results on **WD-LR** to show that the violation rate is small (we sample 10k marginal contributions for each pair of training datum and validation datum):
>
> | Frequency of violations  | **Min** | **25-th percentile** | Median | **75-th percentile** | Max |
> | --- | :-: | :-: | :-: | :-: | :-: |
> | **Across training data** | 1.4e-5 | 1.4e-4 | 2.5e-4 | 3.4e-4 | 1.6e-3 |
> | **Across validation data** | 0 | 4.4e-5 | 1.7e-4 | 4.5e-4 | 1.3e-3 |
>
> This justifies the assumption and validates our theoretical analysis, which explains why NASH leads to significantly better performance than vanilla Data Shapley in Sec. 4.
>
> Next, we will explain how existing assumptions needed to guarantee the effectiveness of top-$m$ Shapley value heuristic are **less realistic**, thus leading to ineffective empirical performances:
> - [1] suggests that Data Shapley works well on monotonically transformed modular (MTM) utility functions. MTM is a much stronger assumption that requires for the ranking of the contribution of data points to be the same across all coalitions. When the MTM assumption does not hold, selecting data points with the top-$m$ highest Data Shapley generally works no better than random.
> In contrast, our consistent player assumption more flexibly allows consistent improvement _or degradation and the ranking can differ_.
> - We also observe that top-$m$ Data Shapley works well only if adding data with higher Shapley values always leads to better total utility on all coalitions. This could be severely unrealistic based on reasons the reviewer has described and our P1 and P2, which causes existing Data Shapley methods to work no better than random.
>
> > It is not clear to me how this would extend to other problems or other utility functions. For instance, how would this work for regression problems where the utility function is not accuracy but something like mean squared error?
> > How would this extend to other problems than classification? For example, regression?
>
> Thank you for the question. Our insights in this paper naturally extend to other problems or utility functions: Complex utility functions involve multiple roles which cannot be captured by a single Shapley value due to P1 and P2 in Sec. 3.1. In contrast, NASH breaks down the utility function to simplier components and accounts for their interaction by nonlinearly aggregating them. For example, NASH would prefer data that reduce the MSE of many poorly predicted validation data instead of well predicted validation data even though the total change in MSE is the same.
>
> To empirically verify this, in Fig. 1 of [our supplementary results](https://anonymous.4open.science/r/nash-rebuttal-E3B0/nash-rebuttal.pdf), we include additional results (1) for regression tasks with negated MSE as utility function; (2) for classification tasks but using validation loss as utility function. The trends are similar to those shown in our paper: Vanilla Data Shapley may work no better than random and our NASH significantly improves over it.
>
>
> > I believe Figure-6 has incorrect labels on the blue line. The semi-values used does not seem to match the labels.
>
> Thank you for spotting this typo and we will correct it in the revision! The labels should be the same as captions of the subfigures.
>
> We hope the above responses and additional experiments improve your opinion of our work. Thank you again for your review and please let us know if you have further questions!
>
> [1] https://arxiv.org/pdf/2405.03875

---

> > ### Author Rebuttal · Reviewer_QwZ2 · 2026-04-03
> >
> > I thank the authors for their detailed rebuttal and addressing my questions regarding assumptions. The authors also addressed the typo mentioned regarding labels. I am satisfied with their response and will increase my score accordingly.

---

> > > ### Author Response · Authors · 2026-04-08
> > >
> > > Dear Reviewer QwZ2,
> > >
> > > We are grateful to hear that our reply has addressed your concerns. We will incorporate the clarifications in our revision. Thank you for your support and engagement during the rebuttal period. We wish you all the best!
> > >
> > > Regards,
> > >
> > > Authors of Submission \#30815

---

### Official Review · Reviewer_PsMn · 2026-03-13

**Soundness:** 4
**Presentation:** 4
**Significance:** 4
**Originality:** 4
**Overall Recommendation:** 6
**Confidence:** 4

**Summary:**

The paper considers the problem of selecting important data points. The starting point of their work is the Shapley value on a boolean function $u_V: 2^{[n]} \to \mathbb{R}$. Here, $u_V(S)$ is the accuracy of a model on the validation set when trained on points in $S$. The key insight of their work is that the different points can be helpful in different ways, and taking the points with the highest Shapley values may inadvertently select points that are helpful in the same ways.

They decompose the value function $u_V$ into an average of value functions $u_v$ where $v$ is a point in the validation set. Here, $u_v(S)=1$ if the model trained on $S$ correctly predicts $1$ and $0$ otherwise. Then,
$$
u_V(S) = \frac1{|V|} \sum_{v \in V} u_v(S).
$$

The paper computes the Shapley values for each one of the $u_v$ functions. (Note: this is about as efficient as computing the Shapley values of $u_V$ because $u_V$ is a linear combination of the $u_v$.) Instead of just averaging/summing the Shapley values of the $u_v$ which would result in the Shapley values of $u_V$ since the Shapley values are linear, the paper uses the Shapley values of the decomposed functions in a non-linear way.

Their work uses the assumption that, for a particular validation point $v$, training points are only ever "good" and "neutral" or "bad" and "neutral". Under this assumption, they show that there is some threshold $\tau_v$ so that, with probability $1-\delta$,
$$
u_v(S) = 1[ \sum_{i \in S} \phi_i(u_v) \geq \tau_v].
$$
Then, with high probability,
$$
\arg \max_{S \subseteq [n]: |S| =m} u_V(S)
= \arg \max_{S \subseteq [n]: |S| =m} \sum_{v \in V} 1[\sum_{i \in S} \phi_i(u_v) \geq \tau_v].
$$
Optimizing this function is computationally hard, so they take the expectation of it (with respect to the $v$, I'm confused here?), and optimize
$$
\arg \max_{S \subseteq [n]: |S| =m} \sum_{v \in V} F(\sum_{i \in S} \phi_i(u_v))
$$
where $F$ is an exponential function with some parameters. They choose $F$ so it is both a) accurate and b) submodular. Hence, a greedy algorithm gives a $1-1/\epsilon$ approximation.

In experiments, they find this method outperforms data Shapley (greedily selecting points according to $\phi_i(u_V)$) and random selections.

**Compliance With Llm Reviewing Policy:**

Affirmed.

**Final Justification:**

My one point of confusion was addressed in the rebuttal

**Key Questions For Authors:**

I'm confused about the expectation in going from Equation 5 to Equation 6. The summation is over validation points $v$ in both. My understanding is that the expectation is over a random draw of $v$. Once $v$ is fixed in the sum, how are you taking the expectation with respect to $v$ inside each term of the sum?

**Limitations:**

Yes.

**Strengths And Weaknesses:**

Strengths:

* Very well written paper, and the story is fantastic: "Why is data Shapley worse than random? Because data points help in different ways, but Shapley values don't capture this. Let's carefully decompose the value function and optimize a submodular objective."

* Persuasive empirical results e.g., Figures 4, 5, and 6! The method looks much better than data Shapley.

* Clever approach to decompose value function and formulate a submodular objective.

I didn't see any major weaknesses. Note: Of the five papers I've reviewed for ICML this year, this is the only one I'm recommending acceptance for and actually think is good. My review is much shorter as a result :)

---

> ### Author Rebuttal · Authors · 2026-03-31
>
> Thank you for the very positive comments regarding our narrative! We’re glad you found the solution clever and the empirical results persuasive. We also thank you for reading our work carefully and writing the comprehensive summary of our work.
>
> > I'm confused about the expectation in going from Equation 5 to Equation 6. The summation is over validation points $v$ in both. My understanding is that the expectation is over a random draw of $v$. Once $v$ is fixed in the sum, how are you taking the expectation with respect to $v$ inside each term of the sum
>
> We would like to clarify that the expectation is not taken over a random draw of $v$. We include each validation point $v$ in the summation to ensure that each validation point/*role* is well-predicted. Instead, we take expectation over threshold $\tau_v$ as it is unknown in Eq. 5. As each threshold is unknown, we consider that it is modelled by some distribution $\mathcal{T}$. On Page 6, we discuss how we choose $F_\mathcal{T}$ (the c.d.f. of $\mathcal{T}$) by connecting it to the shape of learning curves.
>
> The confusion might have arisen because we mentioned in our manuscript that "$\mathcal{T}$ denotes the random variable denoting the threshold value of a *validation datum $v$ drawn uniformly at random from $V$*". We will revise this to *each validation datum $v$* instead.
>
> We hope that this clarification helps and are happy to answer further questions!

---

> > ### Author Rebuttal · Reviewer_PsMn · 2026-04-01
> >
> > My confusion about the relaxation from the indicator (equation 5) to the CDF (equation 6) is resolved.

---

> > > ### Author Response · Authors · 2026-04-08
> > >
> > > Dear Reviewer PsMn,
> > >
> > > We are glad that our response resolved your confusion. Thank you for your recognition of this work and the compliments for our storyline. We highly appreciate your time devoted as a reviwer and we wish you all the best!
> > >
> > > Regards,
> > >
> > > Authors of Submission \#30815

---

### Decision · Program_Chairs · 2026-04-30

**Decision:**

Accept (spotlight)

**Comment:**

Overall, this is a well executed paper with a very solid rationale, an interesting theoretical formulation and convincing results. The reviewers were quite excited by the work and the authors have done a great job responding to reviewer concerns.